# The Current State and 125 kyr History of Permafrost in the Kara Sea Shelf: Modeling Constraints

**Anatoliy Gavrilov** [1,3]**, Vladimir Pavlov** [4]**, Alexandr Fridenberg** [4]**, Mikhail Boldyrev** [5]**, Vanda Khilimonyuk** [1,3]**, Elena Pizhankova** [1,3]**, Sergey Buldovich** [1,3]**, Natalia Kosevich** [1,3]**, Ali Alyautdinov** [2,3]**, Mariia Ogienko** [1,3]**, Alexander Roslyakov** [1,3]**, Maria Cherbunina** [1,3]**, Evgeniy Ospennikov** [1,3]**.**

[1] Lomonosov Moscow State University, Geological faculty, Moscow, Russia

[2] Lomonosov Moscow State University, Geographic faculty, Moscow, Russia

[3] Foundation «National Intellectual Resource», Moscow, Russia

[4] Rosneft Oil Company, Moscow, Russia

[5] LLC «Arctic Research Center», Moscow, Russia

**Abstract**.  The evolution of permafrost in the Kara shelf is reconstructed for the past 125 kyr. The work includes zoning of the shelf according to geological history, compiling sea level and ground temperature scenarios within the distinguished zones, and modeling to evaluate the thickness of permafrost and the distribution of frozen, cooled and thawed deposits. Special attention is given to the scenarios of the evolution of ground temperature in key stages of history that determined the current state of the Kara shelf permafrost zone: characterization of the extensiveness and duration of the existence of the sea during the marine isotope stage MIS -3, the spread of glaciation and dammed basins in MIS-2. The present shelf is divided into areas of continuous, discontinuous to sporadic, and sporadic permafrost. Cooled deposits occur at west and northwest water zone and correspond to areas of MIS-2 glaciation.  Permafrost occurs in the periglacial domain that is a zone of modern sea depth  from 0 to 100 m, adjacent to the continent. The distribution of permafrost is mostly sporadic in the southwest of this zone, while it is mostly continuous  in the northeast. The thickness of permafrost does not exceed 100 m in the southeast , and  ranges from 100 to 300 m in the northeast. Thawed  deposits are confined to the estuaries of  large rivers and the deepwater part of the  St. Anna trench. The modeling results are correlated to the available field data and are presented as geocryological maps. The formation of frozen, cooled and thawed deposits of the region is inferred to depend on the spread of ice sheets, sea level, and duration of shelf freezing and thawing periods.

## 1. Introduction

Permafrost studies and mapping in the Kara Sea shelf (Fig. 1) have a long history. The first evidence of its distribution in the Kara and other Eurasian Arctic shelves appeared in early permafrost maps of the USSR (Parkhomenko, 1937; Baranov, 1960). The distribution  and approximate thickness of the Kara shelf permafrost were first calculated numerically in the early 1970s (Chekhovsky, 1972). By late 1970s - early 1980s, Soloviev and Neizvestnov mapped the whole Russian shelf, and their work became part of the later 1:2 500 000 Geocryological Map of the USSR (Yershov, 1991). The dynamics of subsea permafrost in the Kara shelf during regression and transgression events was reconstructed by modeling (Danilov and Buldovich, 2001). Drilling was first used to study the features of permafrost and its recent formation in the near-shore zone of the Yamal Peninsula (Grigoriev, 1987).

Since 1986, drilling and seismoacoustic surveys have been run by the Arctic Marine Engineering Geological Survey (AMEGS) to constrain the extent of the Kara shelf permafrost and the depths to its top and base; the results were reported in a number of publications (Melnikov and Spesivtsev, 1995; Dlugach and Antonenko, 1996; Bondarev et al., 2001; Rokos et al., 2009; Kulikov and Rokos, 2017; Vasilyev et al., 2018 ). However, the available data are restricted to the southwestern part of the shelf while the more severe northeastern part remains poorly documented and undrilled.

Another recent map of permafrost distribution and thickness in the southwestern Kara shelf (Portnov et al., 2013) is based on seismoacoustic data and modeling with reference to the glacial eustatic curve but without regard to regional features related to the existence of MIS-3 marine terraces. Followed by drilling and seismoacoustic results from the western Yamal Peninsula shelf (Melnikov and Spesivtsev, 1995; Dlugach and Antonenko, 1996; Baulin 2001; Baulin et al., 2005; etc.), geocryological modeling has become quite realistic lately, but its quality is still insufficient for economic activity, even within the best documented southwestern Kara shelf. As for the northeastern and central shelf parts, the knowledge is very preliminary.

The available permafrost maps refer to the isobaths ofthe maximum regression during the peak of cold stage 2 of the marine oxygen isotope stratigraphy (MIS-2). This reference is yet uncertain because the sea depths and level apparently varied in a range of at least tens of meters during the Late Pleistocene-Holocene glaciation history and related isostatic movements, as one may infer from the elevations of MIS-4 and MIS-3 marine terraces in Novaya Zemlya Islands which are

raised high (+45, +55,+60 m, Bolsiyanov et al., 2006) and the adjacent continent (Yamal, Gydan) terraces on which were formed at low sea level (-100 and -70 m respectively, Siddall et al., 2003; 2006)

Given the logistic challenge and high costs of field studies, the knowledge of the Kara shelf permafrost can be extended by numerical modeling. Its application makes it possible to establish the connection of permafrost with components of the natural environment, including glaciations, glacio-isostatic movements and fluctuations in sea level.

## 2. Research methodology and its implementation

       Permafrost in the Arctic shelf is mostly of relict origin: it formed during regressions and cold climate events and
then degraded during Late Pleistocene-Holocene transgressions.

       The methods for subsea permafrost research have been developed since the 1970s and use the retrospective approach of reconstructing the permafrost evolution (Gavrilov, 2008; Romanovsky and Tumskoi, 2011). The history of methods applied to study the structure and t distribution of permafrost of the eastern Russian Arctic was reviewed previously (Gavrilov et al., 2001; Gavrilov, 2008; Nicolsky et al., 2012). We follow these methods in our research and are
trying to extend their work . The work includes compiling a database of paleogeographic, geological, tectonic, and geocryological conditions used further to divide the region according to geological history and for creating possible scenarios of sea level and ground temperature variations that serve as boundary conditions in heat transfer modeling. The general scheme of the methodology is presented in Figure 1.

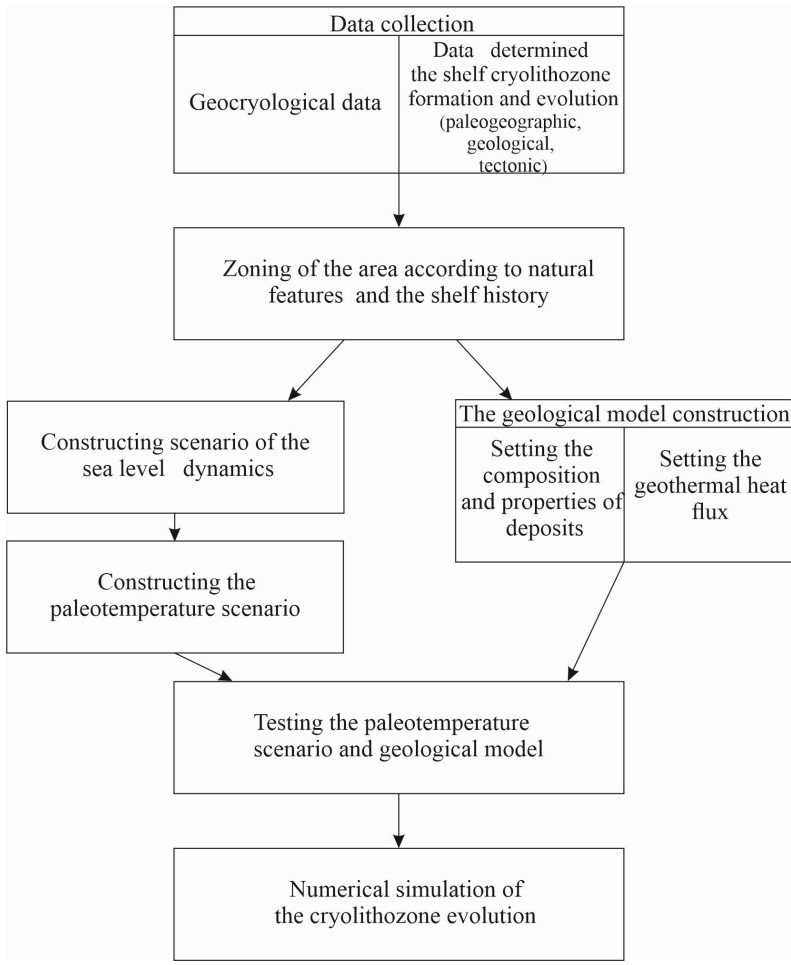

Fig. 1. The general scheme of the methodology.

The zoning is determined by the allocation of territorial units that are characterized by uniformity of formation conditions and, accordingly, by similar (on the given scale of the studies) parameters of permafrost: its distribution, thickness, depths of the top (Kudryavtsev, 1979).The history of subsea permafrost has been modeled using software designed at the Department of Geocryology of the Geological Faculty in the Moscow State University (Khrustalev et al., 1994; Pesotsky, 2016). The software can solve Stefan's problems for non-steady state thermal conductivity assuming moving fronts of pore moisture phase transitions within the modeling domain and variable boundary conditions. The explicit two-layer scheme is applied using the balance method (Pustovoit, G.P, 1999) and the enthalpy formulation of the problem. The code used is publicly available under https://github.com/kriolog/qfrost (last access: 15 February 2020) (kriolog, 2020).

The permafrost dynamics was simulated for numerous paleoclimate scenarios that cover the full range of presumable conditions in the Arctic shelves.. The total number of paleo-scenarios for the Kara sea used in the course of mathematical modeling was 30 .The 1D modeling domain had a vertical size of 5 km to avoid the effect of its base on the permafrost dynamics. The temperatures at the surface according to the available paleoclimate reconstructions and a heat flux at the base of the modeling domain were used as the first- and second order boundary conditions, respectively. The heat flux was assumed to be 50 mW/m2 , which corresponds to the average for most of the shelf territory or 75 mW/m2, as in zones of relatively high heat flux in gas-bearing bottom sediments (Khutorskoy et al., 2013).The modeling was performed for several uniform reference rock and sediment types in order to reduce the number of possible solutions in the conditions of high lithological diversity in the area. Then the modeling results were extrapolated to complex sections that comprise alternating reference lithologies or different combinations of relatively thick layers. We consider two cases to extrapolate our results:In the first case, we consider the uniform alternation of homogeneous layers with a relatively low thickness (relative to the total thickness of the permafrost). The linear interpolation is simply performed in accordance with the percentage of the thickness of frozen ground obtained during modeling for two "pure" soils at a given moment.The second case relates to the two-layer structure of the section, when a sufficiently thick (as compared with the permafrost thickness) homogeneous layer is underlained by a second homogeneous layer of unlimited thickness. Then the following considerations are valid. If the thickness of the upper layer is zero, then the thickness of the permafrost is equal to the result of simulation for "pure" rocks/sediments of the second layer. If the thickness of the upper layer is equal to (or more) the thickness of the permafrost obtained for the first layer, then the problem becomes single-layer and the thickness of the permafrost is equal to that of the first layer. Therefore, when the thickness of the upper layer changes from zero to the thickness of the permafrost of the first layer, the thickness of the two-layer section changes from the thickness of the permafrost of the second layer to the permafrost thickness the first layer. Therefore, as a first approximation, this change is linear (which is not entirely true) and a simple linear interpolation formula can be obtained (Supplement 1)..All deposits were assumed to be saline from top to bottom of the modeling domain .All reference rocks and sediments were considered saline with Ds = 0.8-1.1% according to the concentration of pore saline solution corresponding to that of Kara sea bottom waters concentration (32-34 ‰). The freezing temperature equal to the freezing temperature of sea water ( - 1,8℃) was set for all types of soils for the modeling. This assumption was made due to the fact that all the marine sediments composing the Yamal Peninsula have the close values of the salinity to a depth of 300 m and more . Very high degree of averaging over the properties was used for the modeling caused by the lack of data on the water area. The rare drilling data showed the salinization of sediments through the entire drilling depth. So there was not possible to take into account the salt diffusion, and the salinity did not vary with the depth. Because the modeling has evaluative nature a scheme with complete freezing (thawing) of moisture in the ground at the moving front of phase transitions was used. The content of unfrozen water in the sediments was taken into account by reducing the volumetric heat of phase transitions in the model by the value corresponding to the average content of unfrozen water in different types of rocks at negative temperatures typical of the process under study (Chuvilin et al., 2007). Thermophysical properties, the content of unfrozen water and the heat of

115 the phase transitions of water in the pores, the freezing point of the deposits were set taking into account the indicated salinization. The paleotemperature scenarios and the geological-tectonic model were tested by comparing the present permafrost state estimated by forward modeling with the available field data from well documented areas, to achieve the best fit.The evolution of the shelf permafrost was reconstructed by heat transfer modeling for the Late Pleistocene-Holocene, since 125 kyr, the end of a long-term interglacial transgression. At that time, subsea permafrost had presumably

fully degraded over the whole studied part of the Kara Sea,as the entire north of the West Siberian Plain had been covered by the sea from 140 to 120 Ka (Zastorozhnov et al., 2010; Shishkin et al., 2015) and the temperatures of unfrozen bottom sediments approached the steady state. The modeling results were correlated with field data and both datasets were used for the final geocryological zoning of the Kara shelf region. The Kara shelf has been quite well studied in terms of paleogeography. Figure 2 compiles the various studies available for the paleogeography of this region.

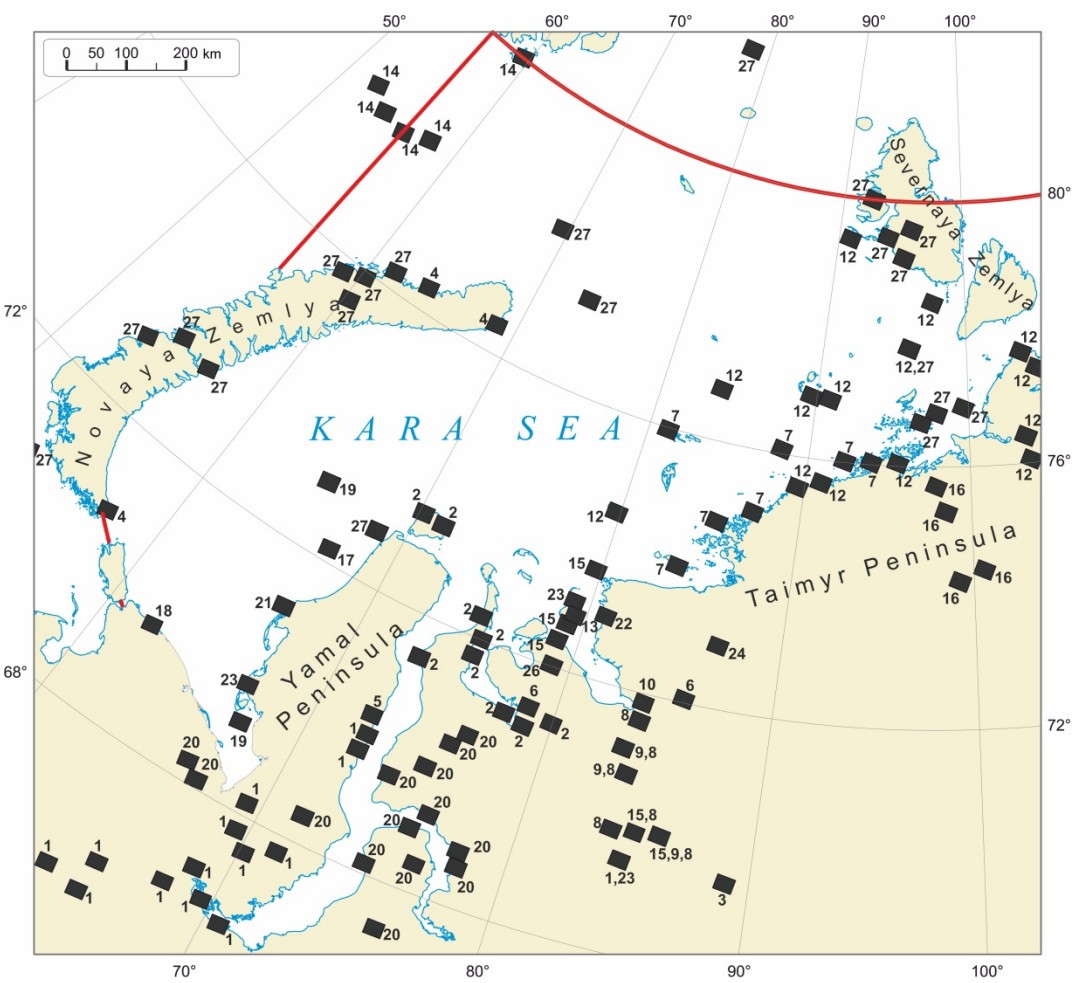

Fig. 2. Late Pleistocene geology of the Kara region: data coverage (the red line is the research region boundary). The

130 numbers on the map correspond to the following publications:

Black squares are sites studied in different years by different research teams. Numbers 1 to 28 refer to publications:
1 = Astakhov and Nazarov (2010); 2 = Baranskaya et al. (2018); 3 = Bolshiyanov et al. (2007); 4 = Bolshiyanov et al. (2009); 5 = Vasil`chuk (1992); 6 = Geinz and Garutt (1964); 7 = Gusev et al. (2016a); 8 = Gusev et al. (2016b); 9 = Gusev et al. (2015a); 10 = Gusev et al. (2015b); 11 = Gusev et al. (2013); 12 = Gusev et al. (2012a); 13 = Gusev and Molod`kov

(2012); 14 = Gusev et al. (2012b); 15 = Gusev et al. (2011); 16 = Derevyagin et al. (1999); 17 = Kulikov and Rokos (2017); 18 = Leibman and Kizyakov (2007); 19 = Melnikov and Spesivtsev (1995); 20 = Nazarov (2011); 21 = Grigoriev

(1987); 22 = Streletskaya et al. (2012); 23 = Streletskaya et al. (2015); 24 = Sulerzhitsky et al. (1995); 25 = Forman et al. (2002); 26 = Gilbert et al. (2007); 27 = Hughes et al. (2016); 28 = Svendsen et al. (2004).

## 3. Paleogeographic scenarios

There is number of ideas about the paleogeography of the Kara region and its development in the Late Pleistocene. Evaluation of the validity of these ideas and the selection of the most reasonable of them was one of the objectives of the present studies. The most popular are two controversial hypotheses implying the presence (Svendsen et al., 2004; Hughes et al., 2016) and absence (Gusev et al., 2012a) of ice sheets in the area. The absence of ice sheets is, however, inconsistent with the existence of (I) Late Pleistocene marine terraces in the Yamal and Gydan Peninsulas and (II) large unfrozen zones in the offshore extensions of major West Siberian rivers (Ob', Yenisei, Taz, and Gyda). The terraces most likely result from a 100 m sea level fall during the Zyryanian cold event (MIS-4). Their origin was possible only by subsidence and uplift due to ice loading (glaciation and low stand) and isostatic rebound (deglaciation and high stand), respectively. The unfrozen zone on the extension of the Ob', Yenisei, Taz, and Gyda river valleys is detectable by drilling and seismic surveys run by AMEGS (Kulikov and Rokos, 2017). In our opinion, it was formed in the place of a freshwater dammed lake (Fig. 3) that existed during the MIS-2 cold event when ice obstructed the continuing river flow. Currently, the largest part of the lake contoured according to the modern bathymetry looks like a shallow-water flat, which area occupies many hundreds of thousands of square kilometers. Within it, only the paleo-valley of the Ob is expressed, it is absent from the Yenisei. The poor expression of the ancient valley network is especially clearly seen when comparing it with that in the eastern sector of the Arctic, where there were no glaciations. Paleovalleys of Khatangi-Anabar, Olenek, Lena, Indigirka, and Kolyma rivers are clearly traced in bathymetry up to the outer shelf. The existence of a dammed lake produced by an ice dam during the MIS-2 cold event is recorded in the estuaries of the West Siberian rivers, which are much longer and farther advanced than those of any other river of the Eurasian Arctic basin. The duration of its existence is explained by the length of the Ob, Taz and Pur estuaries and their flatness. Both of these indicators are close to the record, if not the record for Eurasia: the length of the Ob estuary is 800 km and the average longitudinal slope of its bottom is 1-2 cm/1 km .

Thus, the paleogeographic scenarios used by authors for reference in the modeling of this study assume the existence of ice sheets (Svendsen et al., 2004; Hughes et al., 2016). For the modeling purposes, the shelf is divided by authors into domains, subdomains, areas and subareas according to its 125 kyr history of glaciations and the respective effects on bottom sediments (Fig. 3; Table 1). The largest taxa — the domains— were distinguished by the presence / absence of glaciation and its type during MIS-2, subdomains — by the presence and impact of ice and water cover on bottom deposits in MIS-2. When specifying latitudinal zonality during periods of shelf drying, the authors followed differences in ground temperatures reflected on the Russian Geocryological Map (Yershov ed., 1991). The mean annual ground temperature is 4-6 °C lower in the NE of the region (north of Taimyr) adjacent to the Kara coast, than as for the SW (south-west of Yamal). During the periods of drainage in the periglacial part of the shelf in MIS-2, the following values for the ground temperature were taken: -19 ° C for the north-eastern area, and -15 °C for the south-western one.

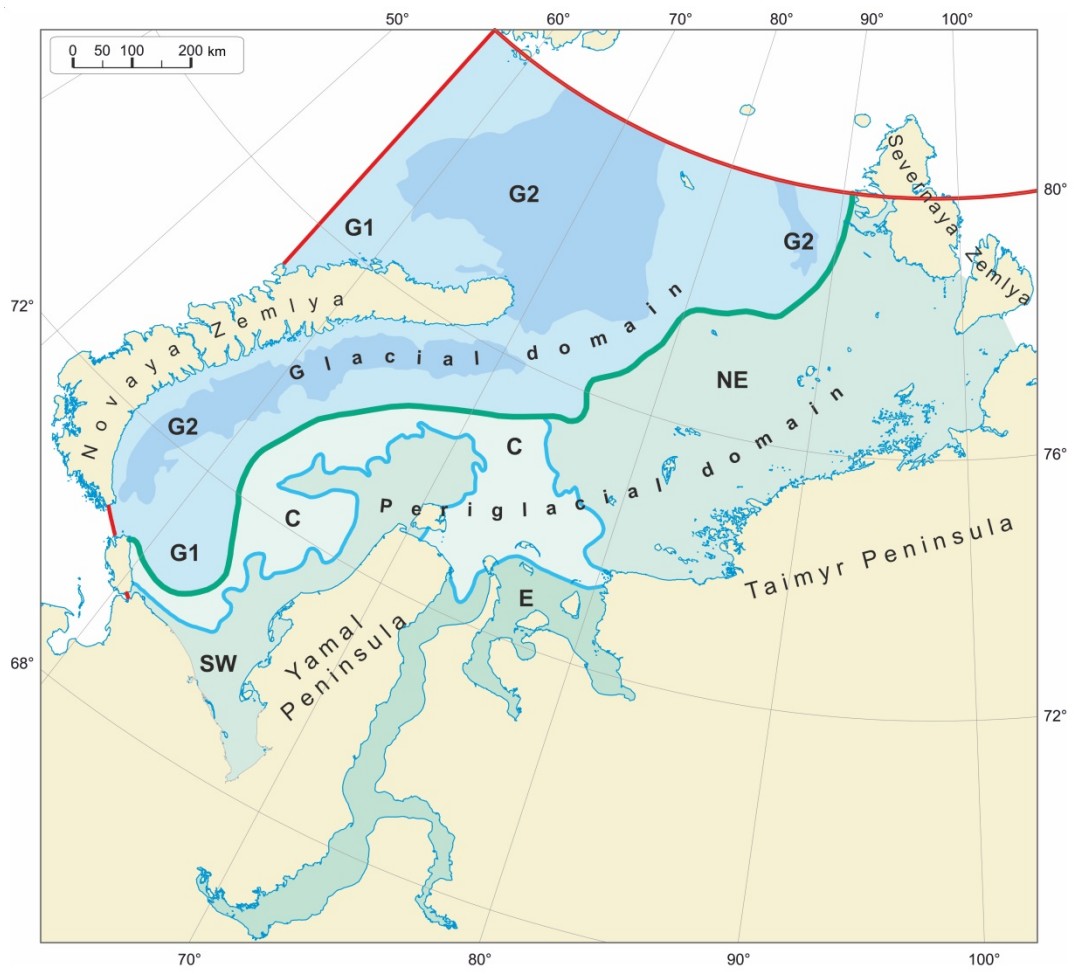

Fig. 3. Zoning of the Kara shelf according to its geological history for 125 kyr. Abbreviations are explained in the text and in Table 1. The red line is the research region boundary

The glacial domain includes zones where the MIS-2 ice sheet reached the sea bottom (G-1) and those of shelf ice at greater sea depths (G-2); the periglacial domain consists of subaerial and subaqual (under the ice-dammed lake) subdomains, which are further divided into areas of the present sea bottom (central shelf, C) and estuaries (E) within the subaqual subdomain and southwestern (SW, 68-71 ° N) and northeastern (NE, 72-77 ° N) shelf parts within the subaerial subdomain (Fig. 3; Table 1). The SW and NE areas had different landscapes during MIS-2 and, correspondingly, differed in ground temperature and freezingdepth. The areas C and E were flooded by cold (<0°C) sea water and warmer (>0°C) river water, respectively, in the Holocene

Table 1. Zoning of the Kara shelf (Late Pleistocene-Holocene history, past 125 kyr, events MIS-2 – MIS-1)

| Domains | Subdomains | Areas (landscapes) | | Subareas |
|---|---|---|---|---|
| Periglacial domain | Subaerial | Southwestern shelf | **SW** | Sea depths 0-120 m (contour intervals of average depths at 5, 20, 50, 80 and 100 m) |
| | | Northeastern shelf | **NE** | |
| | Subaqual (under ice-dammed lake) for 25-15 kyr | Central shelf | **C** | Sea depths 0-80 m |
| | | Estuaries | **E** | |
| Glacial domain, ice reaching sea bottom | Subaqual (for 7-10 kyr) | **G-1** | | Sea depths 0-200 m |
| Glacial domain, shelf ice, MIS-2 | Subglacial-subaqual | **G-2** | | Sea depths 200-800 m |

Finally, the periglacial areas were divided into subareas according to sea depths which controlled the duration of permafrost formation during regressions and degradation during transgressions: 0-10, 10-35, 35-65, 65- 90 and 90-140 m sea depth intervals with average values of 5, 20, 50, 80, and 120 m, respectively.

The subdomains of ice contacting(G-1) or not (G-2) the sea bottom were revealed from seismoacoustic data, with reference to the 1:2 500 000 Map of Quaternary deposits (2010): acoustically transparent deposits (below isobaths 200 m)
were considered as glacial-marine. Glaciers in these places were treated as shelf ice. The zones G-1 and G-2 are only shown in Fig. 3 and Table 1 and were not divided further. The modeling we carried out for the G-1 area showed that, to date, the permafrost formed in the MIS-2 have not survived.

The present permafrost in the region formed under the effect of climate-driven eustatic sea level change. The sea level curve for the 125-15 kyr period was plotted using the eustatic curves of Lambeck and Chappell (2001) and Siddal et al.
(2003, 2006). These curves were adapted to the regional specificity (Trofimov et al., 1975; Streletskaya et al., 2009; Shishkin, M.A. et al., 2015), with regard to Late Pleistocene marine terraces in Yamal interpreted in the context of ice waxing and waning. Special focus by authors was in the recreating on the Late Pleistocene-Holocene transgression (Fig. 4A,B). The Figure 4C shows the scenario on  fluctuations of the sea level during the modeling period of  125-15 kyr..

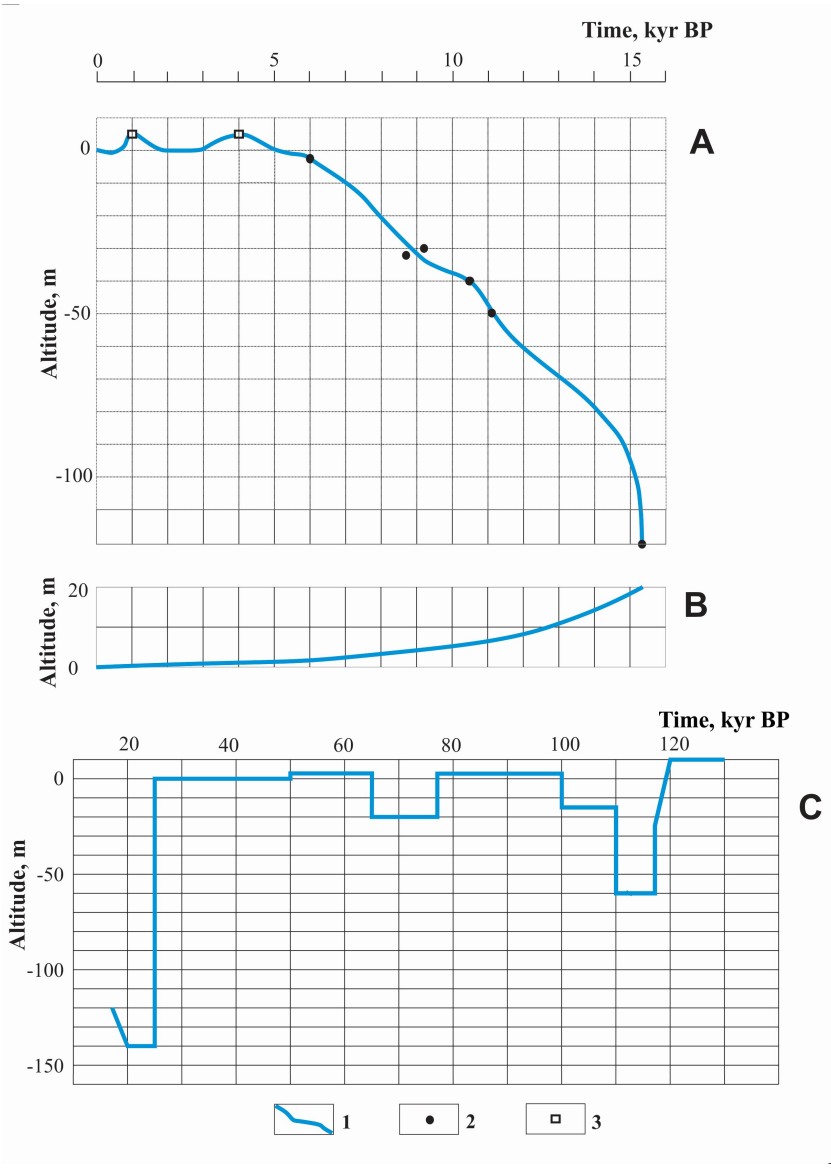

Fig. 4. Scenarios of sea level fluctuations:
for the 15-0 kyr .: A -periglacial domain,  B- glacial domain; for the 125-15 kyr: C - periglacial domain.

1 - sea level curve; 2 - dated bottom sediment cores from Ob' Gulf, Yenisei Gulf, and adjacent offshore (Stein, et al., 2009) and Vilkitsky Strait (Levitan.et al., 2007) sites; 3 - onshore data (Romanenko, 2012);

In the method of constructing a scenario of sea level fluctuations for the Kara shelf, the accounting for glacio-isostatic movements, resulting in the formation of marine terraces (Gutenberg, 1941; Flint, 1957; Bylinsky, 1996), plays a large role. The reason for their formation is the sea flooding of the glacier bed, with a lag reacting to the removal of the glacial load. Therefore, during the reconstructions, post-glacial sea level fluctuations are divided into two categories. In non-glacial areas, sea level rises in relation to the nearest coast, in glacial areas it drops (Lambeck, Chappell, 2001).

In accordance with the above, the scenario is presented in the form of curves for the periglacial (with respect to the continent, Fig. 4A) and glacial (with respect to Novaya Zemlya, Fig. 4B) domains. The periglacial curve was obtained with reference to published evidence on the Laptev Sea (Bauch et al., 2001), and the glacial one was calculated according to data on isostatic subsidence and uplift during glaciation and deglaciation, respectively (Ushakov and Krass, 1972; Nikonov, 1977; Bylinsky, 1996). Figure 4 shows that the calculations for the glacial domain were made taking into account (I) the

thickness of the MIS-2 ice sheet (rising 500 m above the m sea bottom subsidence under the ice load; the fall reached 140 m, or 20 m below the global average regression limit (Lambeck and Chappell, 2001).

     In postglacial times, the sea level rose during transgression in the periglacial domain but fell in the glacial one as a result of Novaya Zemlya uplift. Thus, the glacial domain was exposed to weaker cooling during the glacial period and became flooded and exposed to permafrost degradation right after ice melting. Unlike this, flooding in the periglacial

domain was accompanied by permafrost degradation for as long as 1500 years.

     The construction of scenarios of groundtemperature dynamics by the authors during the estimated time is the final part of the paleogeographic materials for the conduction of the numerical modeling. The ground temperatures were reconstructed in several 1D solutions, with reference to paleo-water chemistry (Fotiev, 1999; Volkov, 2006) and oxygen isotope composition of ice wedges (IW, $\delta^{18}O_{IW}$) (Vasil`chuk, 1992), as well as to reconstructed summer air temperatures.

The $\delta^{18}O_{IW}$ data were corrected according to the results of Golubev et al. (2001).

     We present the paleogeographic scenario as a series of paleotemperature curves adapted to the modeling purposes (Figs. 5, 6). In the following, the plots of mean annual ground temperatures of over the past 125 kyr depending on paleogeographic events (shelf drainage, flooding, glaciation, etc.), and in connection with the existence of latitudinal zoning and meridional sectorality are presented When specifying latitudinal zonality during periods of shelf drying, the

authors followed differences in ground temperatures reflected on the Russian Geocryological Map (Yershov ed., 1991). The mean annual ground temperature is 4-6 °C lower in the NE of the region (north of Taimyr) adjacent to the Kara coast, than as for the SW (south-west of Yamal). When zoning the shelf (Table 1, Fig. 3), the southwest (68-71 ° N) and north-eastern (72-77 ° N) areas were distinguished. During the periods of drainage in the periglacial part of the shelf in MIS-2, the following values for the ground temperature were taken: -19 ° C for the north-eastern area, and -15 °C for the south-

western one .

     The initial conditions are those given for the interglacial MIS-5e . There was a warm-water sea basin on the Kara shelf and adjacent lowlands from 140 to 117 kyr (Fig. 5, 6) (Astakhov and Nazarov, 2010; Nazarov, 2011; Gusev et al., 2016a). Preliminary modeling for the interval MIS-6 - MIS-5e (200-117 kyr BP) showed that previously formed permafrost completely thawed under the sea during MIS-5e, which had existed for more than 20 kyr.The sequence of paleogeographic

events in the form of a series of cartographic schemes is shown in Fig. 7.

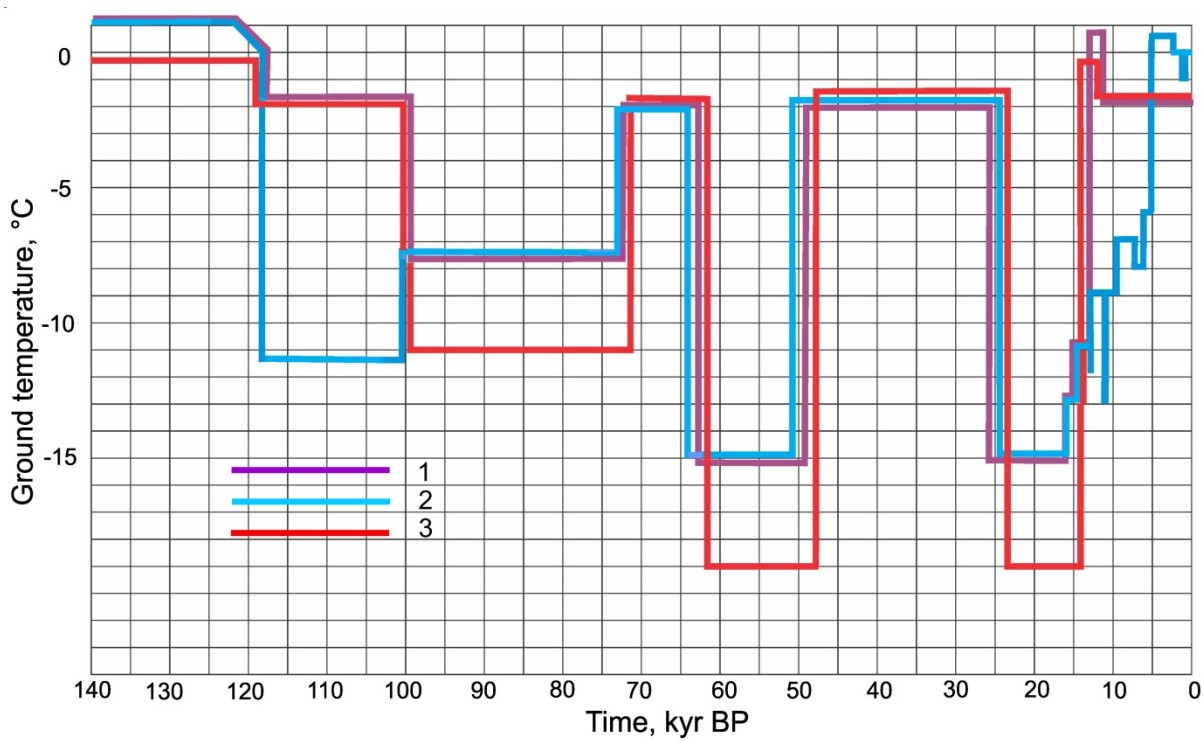

Fig. 5. Paleo-temperature curves for the periglacial subaerial part of the shelf:
1 = southwestern shelf part (SW), 80 m isobath; 2 = southwestern shelf part (SW), 5 m isobath;
3 = northeastern shelf part (NE), 120 m isobath.

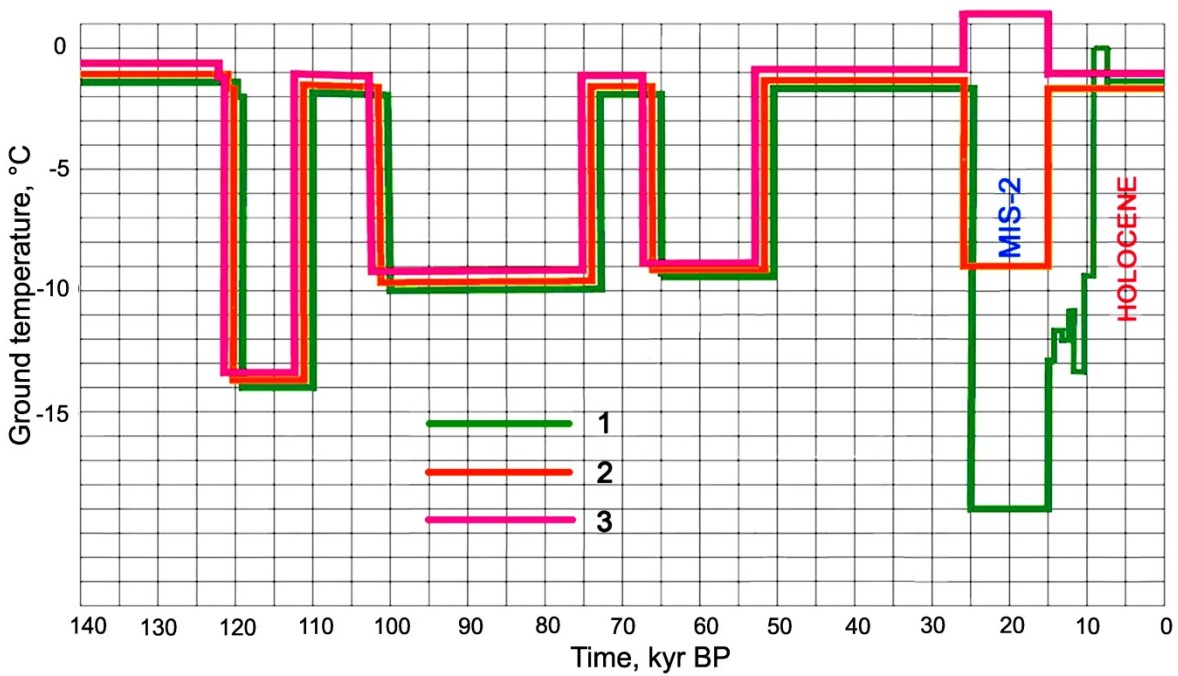

Fig.6. Paleo-temperature curves for 50 m isobath :
1 = periglacial subaerial northeastern shelf part (**NE**); 2 = ice that reached the bottom for 7-10 kyr, **G-1**); 3 = subaqual
(beneath damlake) central shelf part, **C**;

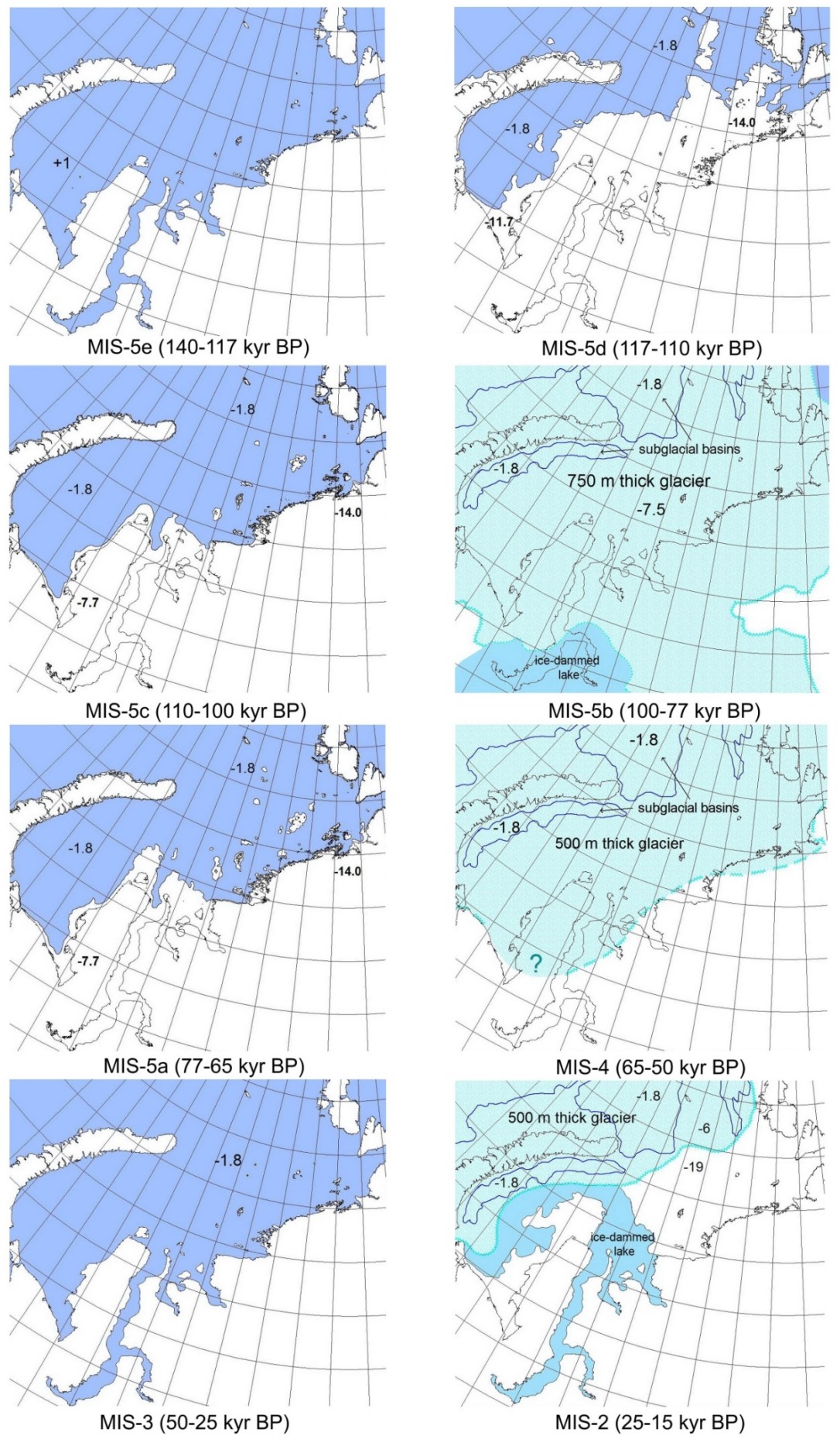


Fig. 7. The series of cartographic schemes illustrating the change in surface conditions in the Kara Sea shelf for the following moments : MIS 5e (140-117 kyr);  MIS 5d (117-110 kyr); MIS 5c (110-100 kyr); MIS 5b (100-77 kyr); 5a (77-65 kyr); MIS -4 (65-50 kyr); MIS -3 (50-25 kyr); MIS -2 (25-15 kyr).

Altogether thirty curves have been obtained, each based on four lithological patterns and two heat flux values.  An important role in the formation of the current permafrost is the last (Holocene) transgression of the sea. Therefore, it was set that that each shelf part stayed for 400-2000 years in the coastal zone where bottom sediments were flooded with saline and warm near-bottom water. It is known that at sea depths from 2 to 7 m in the 1970s (Zhigarev, 1981), up to 10 m in the 2000s. (Dmitrenko) the mean annual temperature of bottom waters in the Laptev Sea stays positive and bottom sediments

thaw from above. For the Kara Sea, there are no such data on isobath intervals, but it is known that temperatures are generally lower. Therefore, we assumed that the interval of isobaths with such water temperatures was limited to 5-6 m. Episodes with such water temperatures occurred on the shelf during the postglacial transgression even in its initial periods, since the July temperature is reconstructed for pre-Holocene warming (allered) exceeding  2 ° C  the temperature of the 1980s. (Velichko et al., 2000). Therefore, we constructed scenarios for these episodes. To determine their duration, dated

data on the absolute altitudes of the sea level during the transgression of the Laptev Sea were used (Bauch et al., 2001). The transgression rate was not the same. The shortest episodes (400 and 375 years) of positive temperatures are determined for periods 15-11 and 10-9 ka respectively. The longest intervals were 11–10, 9–5, and 5–0 Ka with 1000, 750 and >5000 years according to simple calculations. Special mathematical modeling for desalinated  coastal zones was not performed due to the relatively small area of their distribution  and the  specific nature of  salinity distribution over  the water column.

Desalination of waters was taken into account directly when constructing a geocryological map.

The scenarios record alternated cold and warm events accompanied, respectively, by regressions and transgressions (Figs. 5, 6, 7), which created conditions for permafrost growth and degradation. At glaciation peaks, the present sea bottom in the periglacial shelf part was above the shores which subsided under the ice load. This led to the formation of marine terraces, i.e., the sea level fall during cold events (MIS-5b, MIS-4) was smaller than the global

average.

The available data, though far incomplete, show that the sea level between 50 and 25 kyr BP(almost all MIS-3 through) was the same as at present (Fig. 8), in our opinion it is also due to post-Zyryanian uplift (isostatic rebound). On the other hand, the current state of permafrost has been controlled by the ground temperature in the periglacial shelf part and by the presence of an ice-dammed freshwater basin (Fig. 6). In the Holocene, the effect of >0 °C bottom water in the

near-shore zone during warm climate events was critical for permafrost degradation from above (Figs. 5,6). The MIS-2 ground temperature in the glacial shelf part was warmer (Fig. 6, the curve 2) due to thermal insulation by ice.

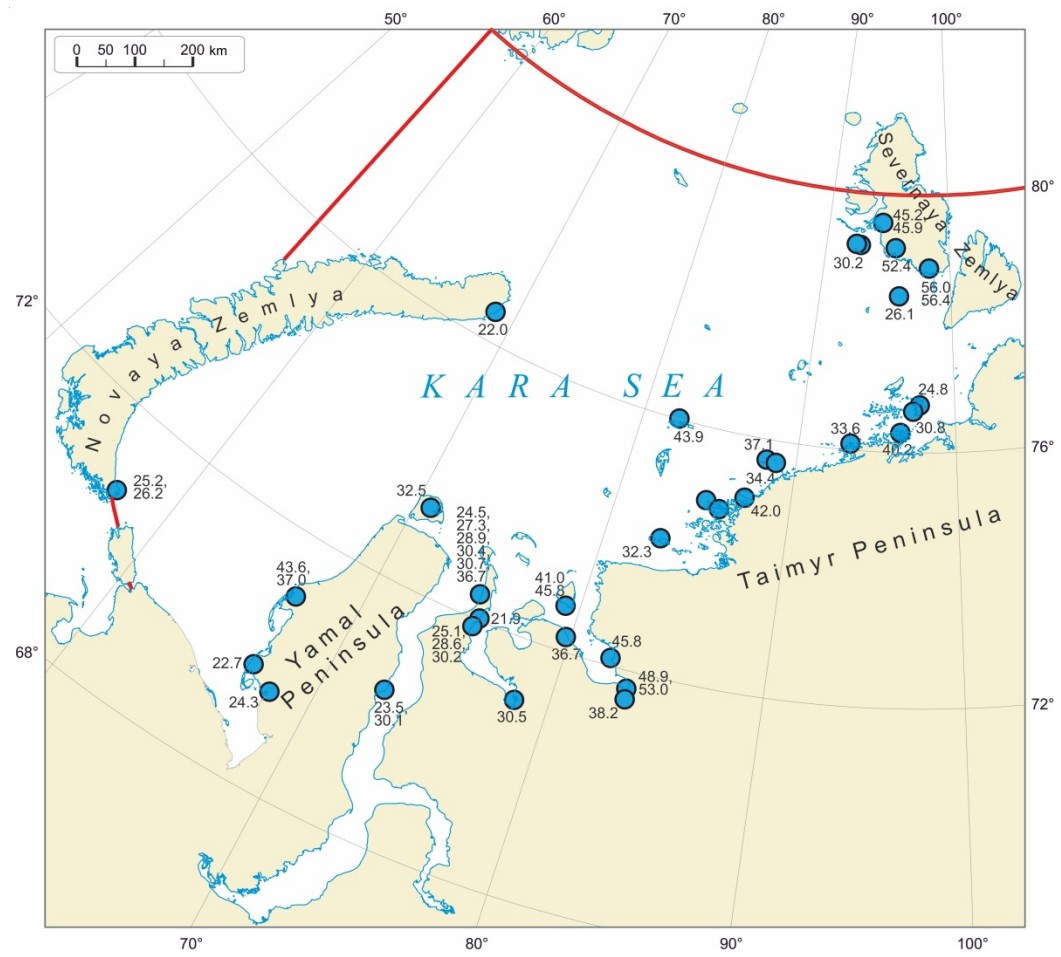

Fig. 8. Location of onshore and offshore sections (points) with marine sediments ([14]C - figures, thousand years ago) of MIS-3 in the Kara Sea shelf, after Gusev et al. (2011; 2012a, b; 2013a, b; 2016 a, b); Baranskaya et al. (2016); Vasil`chuk et al. (1984); Molodkov et al. (1987); Bolshiyanov et al. (2009). The red line is the research region boundary

## 4. Simulation results and regional interpretation

The simulation results show that the distribution of depositsthat differ in their state (thawed, cooled, frozen) are associated with the paleogeographic events of the Late Pleistocene-Holocene. In the distribution of permafrost, a particularly close relationship occurs with glaciation in the MIS-2, the postglacial epoch and the Holocene optimum.

### 4.1. Distribution of frozen, cooed, and thawed deposits

Here we consider the permafrost table (frozen permafrost) corresponds to the -1.8° C isotherm, Therefore further in the text we mean frozen permafrost when talking about «permafrost» and the cryotic unfrozen deposits are «cooled» deposits (marine cryopeg). Permafrost occurs within the areas that were free from MIS-2 ice sheets (SW, NE, C and partly E in Figs. 3,9 ). The present areas of cooled ground were covered with ice that reached the sea bottom during MIS-2 (Fig. 3). Thawed deposits occupy the areas which were open to the inflow of >0 °C Atlantic waters (Levitan et al., 2009) in early postgalacial time (16-15 kyrBP): a part of the G-2 zone (Fig. 3) and a large part of estuaries (E in Fig. 3), except for their northern ends.

Frozen ground is restricted to the periglacial domain. The boundary between the periglacial and glacial domains and, correspondingly, between the frozen, cooled and thawed deposits is delineated by the limits of MIS-2 ice that reached the sea bottom and remained in contact with it for 7-10 thousands of years (Fig. 3), and by the 120 m isobath in the northeastern shelf part, within Severnaya Zemlya and the Cheluskin Peninsula (Figs. 3, 9).

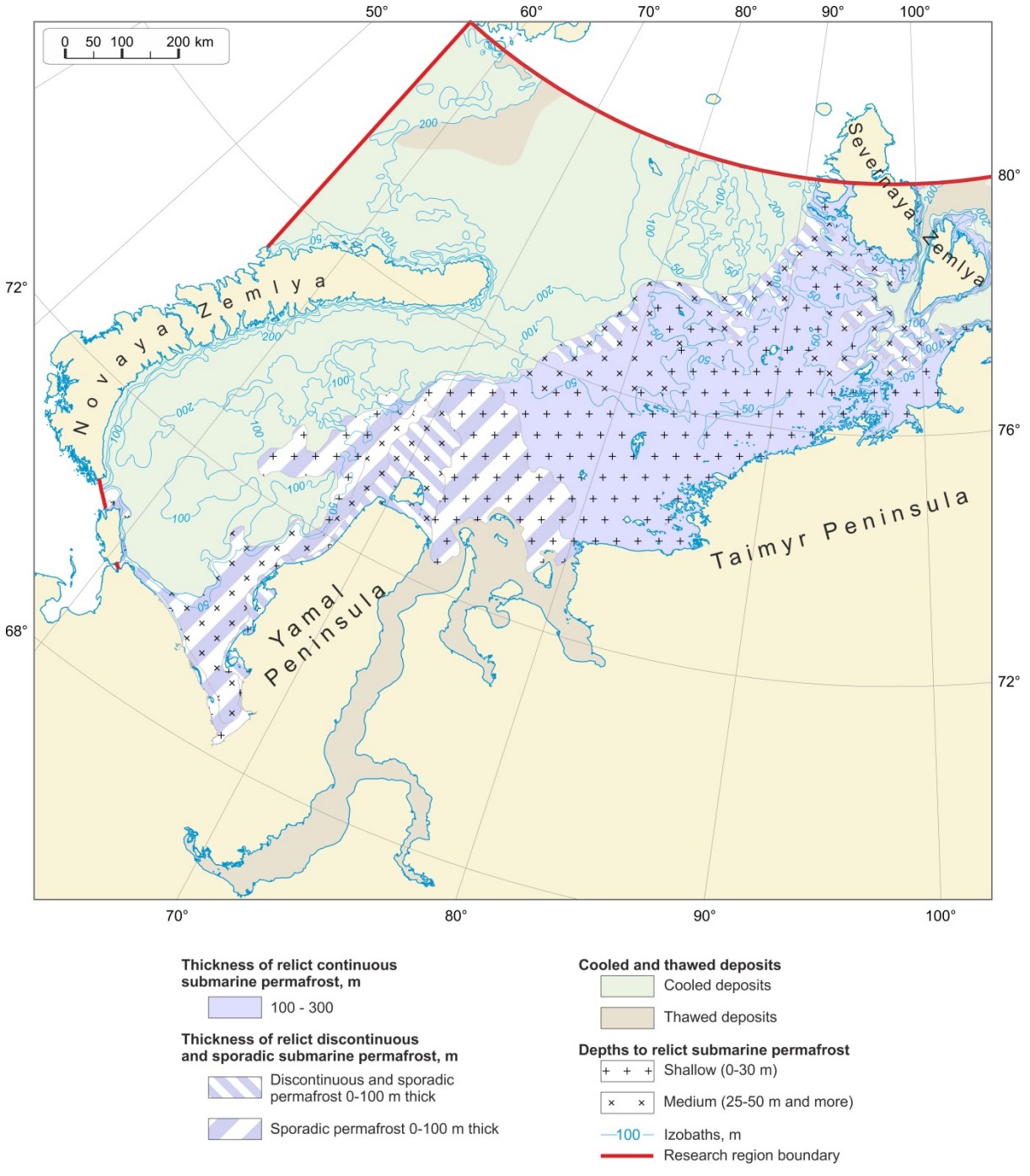

Fig. 9. Fragment of the geocryologic map of the Kara Sea and its legend.

**4.2. Distribution of permafrost, its thickness and depth to the permafrost table**

Permafrost becomes more extensive, shallower and thicker from southwest to northeast and from deep offshore toward near-shore shallow waters. This pattern has several controls (Supplementary Figure 1):

(1) latitudinal climatic zonation and division into sectors;

(2) sea depths that affect the duration of shelf drying and flooding (permafrost formation and degradation, respectively);

(3) ice-dammed freshwater basin in MIS-2;

(4) geothermal heat flux;

(5) lithology and properties of deposits;

(6) sea water and seasonal ice cover (salinity and freezing-thawing temperatures of deposits);

(7) thermal effect of river waters;

(8) Holocene climate optimum.

The southwestern part of the Kara Sea experienced an impact of warm water inputs (Pogodina, 2009) in the middle Holocene (the Holocene optimum), judging by the distribution of thermophile foraminifera communities with predominant Arctic-boreal species found currently in the Pechora Sea. Therefore, the >0 °C bottom water temperatures common to the present-day Pechora Sea may have existed 7-5 kyr in the southwestern Kara shelf and provided thawing of permafrost from above.

The distribution of permafrost also depends on deep heat flux which may reach 50-60 mW/m$^2$ over the greatest part of the Kara shelf (Khutorskoi et al., 2013). Temperature logs from deep boreholes in the shelf and coastal gas reservoirs of Yamal give heat flux values of 73-76 mW/m$^2$. Other controls include lithology, water contents, physical properties, and salinity (freezing-thawing temperatures) of deposits .

Tables 2 and 3 lists the most common depths to the permafrost table determined in previous studies and according to our own calculations. They may vary significantly, even within marine geosystems of the same type: according to drilling results, they are 56 m in the near-shore zone at the Kharasavei Cape and as shallow as 5 m in the Sharapov Shar Gulf 50-60 km in the south. The prospective values are 30-40 m below the sea bottom under the Yamal shores and close to the water table under the retreating coast (Baulin et al., 2005). The permafrost thickness was estimated as the difference between the depths to permafrost table and base

### 4.2.1. Southwestern periglacial shelf

The southwestern part of the Kara shelf (SW in Fig. 3) is occupied by discontinuous and sporadic permafrost (Fig. 8), as inferred from modeling with reference to field data (Dlugach and Antonenko, 1996; Melnikov and Spesivtsev, 1995; Baulin, 2001; Bondarev et al., 2001; Rokos et al., 2001, 2007, 2009; Baulin et al., 2005; Neizvestnov et al., 2005; Kulikov and Rokos, 2017). According to the modeling results, permafrost can exist in sand at a heat flux of 50 mW/m$^2$ but is absent in all other lithologies at 75 mW/m$^2$ (Table 2).

Table 2. Modeling results for depths to permafrost top and base and permafrost thickness, uniform lithology, SW area

| Sea depth, m | Lithology | Geothermal heat flux, mW/m² | | | | | |
|---|---|---|---|---|---|---|---|
| | | 50 | | | 75 | | |
| | | Depth to base, m | Depth to top, m | Thickness, m | Depth to base, m | Depth to top, m | Thickness*, m |
| 5 | sand | 190 | 50 | 140 | 175 | 55 | 120 |
| | clay silt | 110 | 45 | 65 | 0 | | |
| | bedrock | 150 | 45 | 105 | 0 | | |
| 20 | sand | 180 | 50 | 130 | 140 | 60 | 80 |
| | clay silt | 75 | 40 | 35 | 0 | | |
| 50 | sand | 175 | 50 | 125 | 130 | 60 | 70 |
| | clay silt | 0 | | | 0 | | |
| | bedrock | 120 | 60 | 60 | 0 | | |
| 80 | sand | 165 | 65 | 100 | 82 | 55 | 27 |
| 120 | sand | 0 | | | 0 | | |
| | clay silt | 0 | | | 0 | | |
| | bedrock | 0 | | | 0 | | |

*the quoted permafrost thicknesses in all tables here and below are current residual values.

Table 3. Modeling results for depths to permafrost top and base and permafrost thickness, alternated sand and clay silt (clay), SW area

| Sea depth, m | Alternating layers, 0.5 and 0.3 volumetric fraction of sand ($n_n$) | Geothermal heat flux, mW/m$^2$ | | | | | |
| --- | --- | --- | --- | --- | --- | --- | --- |
| | | 50 | | | 75 | | |
| | | Depth to base, m | Depth to top, m | Thickness, m | Depth to base, m | Depth to top, m | Thickness, m |
| 5 | $n_n$=0.5 | 133 | 40 | 93 | 0 | | |
| | $n_n$=0.3 | 117 | 40 | 77 | 0 | | |
| 50 | $n_n$=0.5 | 132 | 50 | 82 | 0 | | |
| | $n_n$=0.3 | 124 | 55 | 69 | 0 | | |
| 120 | $n_n$=0.5 | 0 | | | 0 | | |
| | $n_n$=0.3 | 0 | | | 0 | | |

Table 4. Modeling results for depths to permafrost top and base and permafrost thickness, 50 m of alternated sand and clay silt (clay) lying over bedrock, SW area

| Sea depth, m | Alternating layers, 0.5 and 0.3 volumetric fraction of sand ($n_n$) | Geothermal heat flux, mW/m$^2$ | | | | | |
|---|---|---|---|---|---|---|---|
| | | 50 | | | 75 | | |
| | | Depth to base, m | Depth to top, m | Thickness, m | Depth to base, m | Depth to top, m | Thickness, m |
| 5 | $n_n$=0.5 | 129 | 40 | 89 | 0 | | |
| | $n_n$=0.3 | 115 | 40 | 75 | 0 | | |
| 50 | $n_n$=0.5 | 130 | 55 | 75 | 0 | | |
| | $n_n$=0.3 | 127 | 57 | 70 | 0 | | |
| 120 | $n_n$=0.5 | 0 | | | 0 | | |
| | $n_n$=0.3 | 0 | | | 0 | | |

The quantitative modeling results show (Table 2) that the permafrost state depends on lithology if it is uniform. However, the real shelf sections (e.g., in boreholes of the Leningradskaya and Rusanovskaya fields) consist of different alternating lithologies. Therefore, data listed in Table 2 are applicable uniquely as a check for the role of lithology and properties in the formation, distribution and thickness of permafrost, and in depths to its top and base, which were estimated for layered sections.

Along the west coast of the Yamal Peninsula (area of the Kharasavei-Sea field), where the modern frozen deposits form the upper part of the section, and the relict ones are underlain, permafrost in the Kharasavei shelf gas and condensate deposit forms three elongate zones: ~0.5 km wide zone of continuous permafrost near the shore (0-2 m sea depth); discontinuous permafrost, 1-3 km wide, sea depths to 5-7 m (Baulin, 2001; Baulin et al., 2005); and sporadic permafrost, which may spread to sea depths of 80 m (our estimate), or 100 m (Kulikov and Rokos, 2017) to 105 m (Vasiliev et al., 2018).

The depths to the permafrost table are variable, especially near the shore, as a result of coastal advance and retreat. Permafrost currently exists in zones of coast aggradation, while permafrost beneath retreating coasts is the shallowest at the shoreline and becomes deeper seaward. Specifically, permafrost was stripped at a depth of 0.5 m below the sea bottom at the geocryological site Marre-Sale, in borehole 16-14 drilled for temperature monitoring 0.5 km far from a retreating coast (Dubrovin et al., 2015) but is deeper (3.5 m) in another monitoring borehole 15-14 located 0.9 km far from the shore, where a 0°C mean annual ground temperature was recorded at 3 m subbottom depth. On the other hand, the permafrost table beneath stable coast may be as deep as 30-50 m (Baulin et al., 2005). Similar depth variations occurred also during the Holocene transgression, i.e., this is a typical feature of local permafrost. However, generally the depths to permafrost increase offshore and depend on its composition and ice content, as well as on the time when it submerged.

Within the sea depths from 2 to 5-7 m, the depth to permafrost varies from 5 to 30-40 m. Similar values were inferred by modeling for the most realistic sections of interbedded sand and clay (40 m at a heat flux of 50 mW/m$^2$). The calculated most probable permafrost thickness for these lithologies is 75-90 m at 50 mW/m$^2$. It agrees with 60-80 m reported by Badu (2014) as well as with the estimates 90 m and 70 m by Vasiliev et al. (2018) for the sea depths about 10 m and 35 m, respectively. Permafrost in terrace I of Yamal, which formed from MIS-2 through the Holocene and was not exposed to prolonged subsea degradation, is as thick as 185 m (Trofimov et al., 1975; Yershov, 1989).

According to calculations, on isobaths exceeding 7-10 m, the subsea permafrost table is as deep as 50-80 m (Table 3), which agrees with estimates based on field data (Melnikov and Spesivtsev, 1995). It is most likely beyond the reach of engineering geological drilling (Baulin et al., 2005).

Permafrost is discontinuous to sporadic closer to the shore and sporadic at greater sea depths, with the boundary at the 10 m isobath. The depths to permafrost table are 0-30 in two first zones and 20-50 m in the offshore zone. The permafrost thickness is from 0 to 100 m (Fig. 9).

Similar distribution and parameters of permafrost are common to other parts of the southwestern periglacial shelf, but they vary slightly as a function of zonal features. Farther in the north, the boundary between discontinuous and sporadic permafrost follows the 20 m isobath. The depths to the permafrost table are 30 to 70 m at sea depths from 0 to 20 m according to field data, and 40 to 70 m according to calculations; the calculated permafrost thickness is mainly <100 m (the estimates in Table 2 are for reference sections and those in Tables 3-4 are real data). In real sections within 20-80 m sea depths, permafrost lies at a depth of 50-55 m and is 70-90 m thick (Tables 3-4).

The permafrost changes form discontinuous to sporadic below the 5 m sea depth south of the Kharasavei deposit and is only sporadic still farther to the south (end of the Baidaratskaya Guba Bay).

### *4.2.2. Northeastern periglacial shelf*

There is no drilling data for this shelf part (NE in Fig. 3; Fig. 8). According to logs from a test well in Sverdrup Island, heat flux, varies from 25 to 60 mW/m$^2$ as a function of lithology and thermal properties of rocks and sediments at different core depths (Khutorskoi et al., 2013). Modeling results for a heat flux of 50 mW/m$^2$ predict that the permafrost is continuous at sea depths within 0-80 m and discontinuous to sporadic from 80 to 120 m (Tables 5-7). Cooled rocks may exist at these sea depths mainly in clay sediments, at different heat flux values, though permafrost is predominant (Tables 5-7).

410

Table 5. Modeling results for depths to permafrost top and base and permafrost thickness, uniform lithology, NE area

| Sea depth, m | Lithology | Geothermal heat flux, mW/m$^2$ | | | | | |
|---|---|---|---|---|---|---|---|
| | | 50 | | | 75 | | |
| | | Depth to base, m | Depth to top, m | Thickness, m | Depth to base, m | Depth to top, m | Thickness, m |
| 5 | sand | 270 | 25 | 245 | 175 | 25 | 150 |
| | clay silt | 130 | 20 | 110 | 0 | | |
| | bedrock | 190 | 28 | 162 | 0 | | |
| 20 | sand | 260 | 30 | 230 | 160 | 32 | 128 |
| 50 | sand | 250 | 35 | 215 | 150 | 35 | 115 |
| | clay silt | 120 | 35 | 85 | 0 | | |
| | bedrock | 170 | 40 | 130 | 0 | | |
| 80 | sand | 230 | 40 | 190 | 105 | 40 | 62 |
| 120 | sand | 212 | 50 | 162 | 85 | 50 | 35 |
| | clay silt | 0 | | | 0 | | |
| | bedrock | 90 | 50 | 40 | 0 | | |

Table 6. Modeling results for depths to permafrost top and base and permafrost thickness, alternated sand and clay silt
(clay), NE area

| Sea depth, m | Alternating layers, 0.5 and 0.3 volumetric fraction of sand $(n_n)$ | Geothermal heat flux, mW/m$^2$ | | | | | |
|---|---|---|---|---|---|---|---|
| | | 50 | | | 75 | | |
| | | Depth to base, m | Depth to top, m | Thickness, m | Depth to base, m | Depth to top, m | Thickness, m |
| 5 | $n_n$=0.5 | 221 | 20 | 201 | 107 | 20 | 87 |
| | $n_n$=0.3 | 190 | 20 | 170 | 75 | 20 | 55 |
| 50 | $n_n$=0.5 | 206 | 35 | 171 | 91 | 38 | 53 |
| | $n_n$=0.3 | 171 | 35 | 136 | 61 | 37 | 24 |
| 120 | $n_n$=0.5 | 121 | 45 | 76 | 0 | | |
| | $n_n$=0.3 | 85 | 40 | 45 | 0 | | |

420

Table 7. Modeling results for depths to permafrost top and base and permafrost thickness, 50 m of alternated sand and clay silt (clay) lying over bedrock, NE area

| Sea depth, m | Alternating layers, 0.5 and 0.3 volumetric fraction of sand ($n_n$) | Geothermal heat flux, mW/m$^2$ | | | | | |
|---|---|---|---|---|---|---|---|
| | | 50 | | | 75 | | |
| | | Depth to base, m | Depth to top, m | Thickness, m | Depth to base, m | Depth to top, m | Thickness, m |
| 5 | $n_n$=0.5 | 214 | 21 | 193 | 0 | | |
| | $n_n$=0.3 | 208 | 21 | 187 | 0 | | |
| 50 | $n_n$=0.5 | 195 | 37 | 158 | 0 | | |
| | $n_n$=0.3 | 188 | 36 | 152 | 0 | | |
| 120 | $n_n$=0.5 | 103 | 43 | 60 | 0 | | |
| | $n_n$=0.3 | 95 | 40 | 55 | 0 | | |

425

The depth to the top of continuous permafrost is most often from 20-25 m near the shore to 35-40 m at sea depths 50-80 m (Table 6). The permafrost thickness varies from 150 to 200 m in sand-clay sediments depending on lithology and sea depth. Correspondingly, the map of Fig. 8 shows typical depths to permafrost in a range of 0 to 30 m and permafrost thicknesses from 100 to 200 m. Discontinuous permafrost lies at a depth of 40-45 m and is 0 to 100 m thick.

### 4.2.3. Central area

The central area covers the offshore extension of the Ob', Yenisey, and Gyda rivers (C in Fig. 3; Fig. 9) in the middle between the southwestern shelf part from the northeastern one. The area was interpreted (Kulikov and Rokos, 2017) as an unfrozen zone (a talik) corresponding to paleo-estuaries and paleo-deltas of the West Siberian rivers. In our view, however, it was rather a lake-like freshwater basin that formed when ice dammed the continuing flow of the large Siberian rivers (Ob, Yenisei, Taz, etc.) during the MIS-2 glacial event. The ice dam caused flooding of the respective shelf part in different periods; the surface area and elevations of the lake table (C in Fig. 3) changed accordingly with the ice sheet contours. Therefore, our map delineates the lake by isobaths from 0 to 80 m.

The Central area is shown in the map as occupied by sporadic permafrost: sea bottom rises within its limits may be frozen (like the cases of Yamal and Ob' Gulf). In fact, permafrost patches beneath the Sartan damlake are scarce as the deposits had stored large heat resources while staying under the ice sheet for thousands of years. The sediments currently occurring within the lake limits were much warmer than the surrounding ground during MIS-2: +4 °C against -19 to -23 °C, respectively. The patches of permafrost lie at depths from 0 to 30 m and the permafrost thickness is a few tens of meters.

### 4.2.4. Estuary (bays)

Permafrost in the quite well documented Ob' estuary (E in Fig. 3) is restricted to a narrow strip along the shore; it is relict permafrost beneath the coast exposed to marine erosion. The total thickness of present and relict permafrost exceeds 20 m within the 1 m isobath, is no more than 10 m at sea depths between 0 and 2-3 m, and pinches out seaward in deeper water; the depth to subsea permafrost within 3 m of water depths is 5 to 15 m or more (Baulin, 2001). Coastal permafrost is traceable as far as the Cape Kamenny at 68°N (Kokin and Tsvetinsky, 2013). The patches of frozen ground are limited to a 100 m wide strip along the shore (less often 300 m).

Seismic reflection profiling reveals numerous gas reservoir zones (Rokos and Tarasov, 2007) and presumably a frozen deposits at a sea depth of 15 m at 71° N (Slichenkov et al., 2009). The estuaries (south of 71.5° N) are mapped as mainly thawed zones with near-shore permafrost at ≤3 m sea depths.

We agree with the previous inference (Melnikov et al., 1998; Badu, 2014) that permafrost within gas fields (Rusanovskaya and other gas-bearing geological structures) does not refer to relict frozendeposits. Their genesis and cryogenic age is the subject of debate.

## 5. Conclusions

The present state and 125 -kyr history of permafrost in the Kara shelf has been modeled with reference to the existing knowledge. Since 125 kyr ago, the parameters of permafrost have been controlled by glacial and isostatic rebound events: frozen ground is present currently in places which were free from ice but is absent from those covered by ice during the MIS-2 glacial.

Degradation of glaciation led to ice rebound and related transgression. As a result, permafrost thawed, partially after the MIS-5b cold event and completely after the MIS-4 event (during the Karginian interstadial, MIS-3).

As a result, the present shelf comprises frozen, cooled, and thawed deposits. Permafrost occurs in the periglacial domain, cooled rocks correspond to areas of MIS-2 glaciation, while thaweddeposits occupy the areas that were exposed to

the effect of >0 °C Atlantic waters (e.g., the western St. Anna trench) in the early postglacial time (16-15 kyr BP); thawed
depositsare also found over the overwhelming part of river estuaries in northern West Siberia, except for the river mouths.

The periglacial shelf part is divided into zones of continuous, discontinuous to sporadic, and sporadic permafrost. The geocryological conditions become harsher (continuous, deep, and thick permafrost) from southwest to northeast and from large sea depths to near-shore shallow water.

The distribution and parameters of permafrost have had multiple controls:

(1) latitudinal climatic zonation and division into sectors;

(2) sea depths that affected the duration of regressions and transgressions and the related freezing and thawing of permafrost, respectively;

(3) an ice-dammed freshwater lake that existed during the MIS-2 event;

(4) geothermal heat flux;

(5) lithology and properties of deposits;

(6) sea water and seasonal ice cover that affect salinity (and hence freezing-thawing temperatures) of deposits;

(7) thermal effect from river waters.

The periglacial shelf part is divided into the southwestern, northeastern, central, and estuary areas. The southwestern area comprises two subareas: a zone of discontinuous to sporadic permafrost along the shore, within 20 m sea depths in the north and 5-7 m in the south, which grades seaward into a zone of only sporadic permafrost. In the former zone, the permafrost has its table at depths from 0 to 30 m and is a few tens of meters to 100 m thick; in the latter zones, the depth to permafrost is 20 to 50 m and the permafrost thickness is less than 50 m.

In the northeastern area, continuous permafrost occurs within sea depths from 0 to 80 m and has a thickness of 100-200 m; the depth to its top varies from 0 to 30 m. The 80-120 m sea depth interval is occupied by ≤100 m thick discontinuous to sporadic permafrost with its table at a depth of 20-50 m.

Permafrost in the central area is sporadic; it is within 50 m thick and its top lies from 0 to 30 m deep. Deposits in the estuaries are mostly unfrozen; relict permafrost is restricted to a 100-300 m strip along the shore.

The studies performed were based on drilling and seismic acoustic data published to date. The study of the shelf by drilling and geophysical methods continues. Therefore, the results of the studies performed by the authors can be used in planning new drilling and in the geocryological interpretation of seismoacoustic profiling.

## Acknowledgement

The authors are grateful to all project participants in the for assistance in collecting and processing an extensive amount of information from various fields of knowledge that was necessary for conducting the research and obtaining the presented results. The project was carried out by the Foundation «National Intellectual Resource» (involving experts from Moscow State University) jointly with the Arctic Research Center of the Rosneft Oil Company.

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
