# Peer review of "The Current State and 125 kyr History of Permafrost in the Kara Sea Shelf: Modeling Constraints"

_The Cryosphere, 2019_

## Referee Comment (RC1) · Anonymous Referee #1 · 10 Dec 2019

Review of The Current State and 125 Kyr History of Permafrost in the Kara Sea Shelf: Modeling Constraints by Anatoliy Gavrilov, Vladimir Pavlov, Alexandr Fridenberg, Mikhail Boldyrev, Vanda Khilimonyuk, Elena Pizhankova, Sergey Buldovich, Natalia Kosevich, Ali Alyautdinov, Mariia Ogienko, Alexander Roslyakov, Maria Cherbunina, and Evgeniy Ospennikov for The Cryosphere

1. Does the paper address relevant scientific questions within the scope of TC? Yes, the focus of the paper is offshore permafrost, an important and understudied component of the cryosphere. The central thrust of the paper is to increase our knowledge of the distribution of permafrost below the Kara Sea using the state of the art knowledge

of regional geology, and ice sheet and sea level histories.

2. Does the paper present novel concepts, ideas, tools, or data? Yes, the study uses a novel mix of physical numerical modelling with specific regional knowledge of processes that influence permafrost.

3. Are substantial conclusions reached? Yes, a new map of the Kara Sea region subsea permafrost is generated. The methods used are somewhat novel in terms of how geological information is combined with palaeohistorical reconstructions. The resulting permafrost distribution is substantial as a contribution to the relatively limited body of literature on Kara Sea permafrost and has implications for the distribution of subsea permafrost elsewhere in the Arctic.

4. Are the scientific methods and assumptions valid and clearly outlined? More detail on the numerical methods used to obtain the presented results needs to be provided.

5. Are the results sufficient to support the interpretations and conclusions? Yes.

6. Is the description of experiments and calculations sufficiently complete and precise to allow their reproduction by fellow scientists (traceability of results)? No, as stated above, the model needs better description. Many journals are moving towards requiring provision of the numerical model used to obtain results; it is to be expected that these need to be made accessible in the future. This is not a requirement for publication in The Cryosphere. Nonetheless, the description of the physics behind the model should be explicit enough to allow reproduction.

7. Do the authors give proper credit to related work and clearly indicate their own new/original contribution? Yes.

8. Does the title clearly reflect the contents of the paper? Yes.

9. Does the abstract provide a concise and complete summary? Concise: yes; complete: no. See comment below.
10. Is the overall presentation well structured and clear? Yes.

11. Is the language fluent and precise? The language is fluent but needs improvement in precision and word choice.

12. Are mathematical formulae, symbols, abbreviations, and units correctly defined and used? Yes.

13. Should any parts of the paper (text, formulae, figures, tables) be clarified, reduced, combined, or eliminated? No.

14. Are the number and quality of references appropriate? See comment below.

15. Is the amount and quality of supplementary material appropriate? Yes.

This paper is a timely and well-thought out contribution using a modelling approach to estimate subsea permafrost distribution in the Kara Sea. The authors bring their substantial knowledge of the literature, especially Russian, and their expertise in the palaeohistory and geology of the region to improve the results of their numerical modelling by imposing a hierarchy of geographic regions. This approach and the detailed background work is a significant contribution to the body of knowledge on the topic and will be a paper to which future work in the region will be referred. For these reasons, I recommend that it be published.

I do have one major criticism and a host of suggestions to improve the paper and make it accessible to a wider readership. Although these suggestions may sound overly negative, please accept them in the spirit in which they are made: in the hopes that some of them may lead to improvement of the paper.

General comments: 1. My main criticism of the paper is that the model used has not been adequately explained. It would not be possible for someone using the same or a different model to reproduce the work without a great deal of extra information. In my specific comments, I try to identify and ask questions at points in the text where vague language is used, or where important details are omitted. To allow comparison of their

work, I encourage the authors to be as explicit and detailed as possible about how they achieved their results. Examples of open questions regarding how the model works are: - Line70: what is meant by "double-layer" explicit solution? Is the Qfrost software publically available (e.g. on GitHub) – if so, why not reference it? If the model is well-documented, this might be a sufficient means of answering many of the questions regarding the method. - Line 80: "extrapolated" – what method was used to extrapolate results from monolithic stratigraphies to more complex stratigraphies? Introducing changes in porosity, grain size, thermal properties, etc. would presumably change the temperature field solutions? - On line 82, all rocks (please replace) were assumed to be saline – why? how saline was the sediment assumed to be? Did salinities vary with depth? Was salt diffusion permitted? How did salt content affect the freezing characteristic curve or liquid water of frozen material? - How are discontinuities avoided at the borders between domains/subdomains/areas/subareas?

As a result, claims are made in the paper, but there is not enough information given to the reader to be able to judge whether the claim is justified or on what basis it has been made. For example: - section 4.2 lists 8 controls on the "pattern of permafrost distribution", but 2 of the 8 (lithology and properties or rocks, and Holocene climate optimum) are not described in any detail, making it impossible for the reader to follow the argument or design studies that reproduce the work. A 3rd control (thermal effect of river waters) is not even modelled, so it is not clear how the authors can conclude that this acts as a control. It seems to be an assumption in the model design, but not enough information is provided for the reader to be able to judge. - Tables 2-7 lists the model output for "depths to permafrost top" – but what is merant by "permafrost top" has not been defined anywhere. Does this correspond to an isotherm, the presence of any ice, or of some minimum amount of ice? Or does the model output the depth of the phase change boundary?

And finally, also related to a better description of the model: The discussions and conclusions would be strengthened if the authors included a section that takes a critical

look at how their choice of model, its parameterization and its sensitivity to parameter choice affected the outcome of their study. This is valuable on two levels: it provides a measure for how robust the conclusions of the study are, and it provides a basis for evaluating the sensitivity of the Kara Sea system to changing boundary conditions, allowing some indication of the system's sensitivity to future changes.

3. The language used in the paper is sometimes imprecise or even incorrect; the paper should be proof-read by an native-speaker with some background in the topic. As examples: a. "cold": is probably being used to refer to cryotic conditions, or to conditions below the freezing point. As it stands, it is a vague descriptor. b. "rock": is used to refer to earth material, including either rock or sediment, consolidated or unconsolidated material. In English, "rock" is used to refer only to bedrock material, and would exclude sedimentary deposits of terrestrial, marine or other origin. As it stands, all instances of the use of "rock" need to be replaced with something more precise. c. More examples are given below in the specific comments.

4. The abstract is extremely short and does not provide enough information for a reader to decide whether he/she wants to read the paper. It needs to introduce the larger context for the study, the central question/focus/hypothesis, more detail on the method. It should report key results, findings and conclusions, and may suggest implications or outlook based on the study.

5. The reference list is incomplete. Vasiliev & Rekant (2018) are missing, for example. The reference list needs to be cross-checked with the submitted paper. Some reference citations still include initials (see Fig. 4 caption, for example).

6. This paper stays true to the general phenonmenon of Russian authors citing mostly Russian work, and Canadian/Alaskan researchers citing mostly North American. For citations dealing with regionally specific processes, this is understandable. But neglecting to look at how the North American community has approached modelling the exact same processes under different conditions is harmful in two ways: it exposes the

work to the criticism of being too narrow in its approach, and it makes it less likely that the work will be found and cited by North Americans. I encourage the authors to show their familiarity with the field by referring to the work of researchers from outside of their region, who have presented novel ideas in the field of subsea permafrost modelling. Some examples: - Whitehouse, P. L., Allen, M. B., Milne, G. A. Glacial isostatic adjustment as a control on coastal processes: an example from the Siberian Arctic, Geology, 35, 747–750, doi:10.1130/G23437A.1, 2007. - anything from the group of Romanovsky and Nicholsky (e.g. Nicolsky, D., Shakhova, N. Modeling sub-sea permafrost in the East Siberian Arctic Shelf: the Dmitry Laptev Strait. Environmental Research Letters, 5(1), 15006, 2010.). - anything from Taylor, A. (e.g. Taylor, A. E., S. R. Dallimore, P. R. Hill, D. R. Issler, S. Blasco, Wright, F. Numerical model of the geothermal regime on the Beaufort Shelf, Arctic Canada since the Last Interglacial, J. Geophys. Res. Earth Surf. , 118, doi:10.1002/2013JF002859, 2013.).

Specific comments: Line 13: the use of Kyr as a unit does not follow SI. Line 24: "In the latest . . . earliest . . ." needs correction. Line 45: "raised high" – please quantify Line 49: add "and" and remove "and so on" Line 58: replace "provide their progress" with "extend their work"? Line 59: what is meant with "geocryological results"? Please specify. Line 61: "obtained estimates" – of what? Please specify.

Fig. 1. This figure provides an overview of the method, but uses many general or non-specific terms that reduce the amount of information communicated: - in the top box, what is meant by "environmental data"? - in the second box, what is meant by "conditions"? - in the left third box, delete "dynamics" (adds nothing to "history") - in the third right box & in the fourth left box, replace "rocks" - in the fourth right box: "density of the heat flow from the depths" is usually referred to "geothermal heat flux" - in the fifth box: "Testing . . . of the model" is almost entirely free of content. How was which model tested? More specific word choice could make this box informative - the lowest box is actually two steps: "coordination" and "mapping" - what is meant by "coordination"? This question is never really answered in the paper, but is critical for understanding

what was done. Does the model output get changed in some way by comparison with field data? Where and where not? How? These are important points for anyone wanting to reproduce or apply the method in the same or in other geographical regions.

Line 72: "Permafrost dynamics were..." Line 73: "including..." suggests that other scenarios/conditions were NOT included? How many and why not? Line 77: how were regions of different geothermal heat flux mapped or determined? Line 79 and in all following text: modelling was probably not restricted to the "rock". Line 87: "subsea permafrost had presumably fully degraded..." – this statement requires a reference, especially in light of modelling, for example by Romanovsky, N. N.), showing permafrost elsewhere persisting through interglacials; this point is important, since other researchers have shown that a systematic bias in model results is obtained depending on the initial conditions. Such results show that setting permafrost to zero at the interglacial will introduce a warm bias, that at least would need to be tested.

Fig. 2: it looks like only 14 sites out of more than 100 are located on the shelf, i.e. pertain to subsea permafrost. Is this correct? Please add a description of the red line (which is currently not described until Fig. 8).

Line 106: "the existence of a number of idea about its development..." is not a peculiarity of any region, it is true of every region! Line 115: dammed lakes are invoked to explain the unfrozen zone. Why is the sensible and convective heat transport at the river bed and in the estuarine regions not sufficient to explain the absence of frozen material? Surely the rivers maintain and have maintained positive benthic temperatures for long periods? Line 119: "Insignificance of the severity" is convoluted language that should be simplified. Line 154: explain the abbreviation "MMP" Line 156: "sea level" rather than "sealevel" Line 201: on what basis was it decided how long each portion of the shelf spent in the coastal zone (400-2000 years)? Why were waters in this zone saline? – is this not the zone most affected by the freshwater layer above the halocline, by snow melt and river runoff? Dmitrenko et al (2011) show the freshwater nature of the coastal zone in the Laptev Sea. And why was this zone warmer? Bedfast ice can

result in cooling of the seabed from 0 – 2 m water depth. A little more justification and specification of these boundary conditions, which determine the most immediate and rapid response of permafrost to inundation by seawater, are necessary.

Fig. 8: The map shows permafrost thickness, which can clearly result as output from 1D numerical modelling. What conditions were applied to determine zonation of permafrost based on continuity (continuous, discontinuous, sporadic)? I.e. how do conclusions about distribution result from 1D modelling? Caption: why are only "fragments" of the map shown? Why not present the reader with the whole map?

---

## Referee Comment (RC2) · Anonymous Referee #2 · 14 Jan 2020

It is a very interesting and timely publication, considering various efforts to develop a circumpolar map of the subsea permafrost in the Arctic.

One of the main accomplishments of this manuscript is development of the up-to-date paleo-geographic scenario for the air temperature, sea level change, assumptions about the glaciation, etc. At the same time, it is hard to duplicate the employed paleo-geographic scenario. Here are my suggestions: 1. Plot the air temperature (ground surface temperature) for different regions for the last 125k years. As I understand there is a temperature zonation factor involved. What is it? How does the air temperature differ between various regions, subzones? 2. It is stated that "about 50-75ky bp the seal

level was the same as present". Unfortunately, the paper lacks the sea level dynamics, I think it was used in the scenario building, but not explicitly shown. Please supplement Figure 4 with a plot of the relative sea level curve with respect to the present-day conditions for the last 125k years. 3. Make a series of maps to show which areas were flooded, glaciated or exposed to air 10, 20, 40, 55, 70, 80 ky bp. It will help a reader. It is okay to take different times, e.g. middle of the MIS periods. The goal is to help future researchers to understand when a certain part of the shelf was exposed to the air. Right now, it is not clear. Also, change the y-labels in Figure 4 to "Sea level with respect to the present-day datum, m". Make values in Figure 4a negative. 4. How are the effects of salts taken into the account? Does the salt lower the freezing point depression? Do you take into the account unfrozen liquid pore water while freezing the saline water? How are the salt effects parameterized in the model? Explain and clarify in the paper. 5. The paper goes into the discussion of various ground layers (sand, clay, etc) and heat flux values. I suggest moving this analysis into a separate section, e.g. "Sensitivity analysis". Do you do sensitivity analysis with respect to salt concentration? 6. The manuscript is understandable, but terminology is used unconventionally.

Thank you and good luck.

---

## Author Comment (AC1) · 14 Jan 2020

Dear referee, thank you very much for your review. It has helped us to substantially improve the manuscript. We moderated our language focusing on precision and word choice. We expanded the abstract by adding specific results on the distribution and thickness of permafrost on the shelf. As for the model, the detailed description of the model used will be added.

Please find detailed responses to each of your comments by the link:

Please also note the supplement to this comment:

[Figure]

https://www.the-cryosphere-discuss.net/tc-2019-112/tc-2019-112-AC1-supplement.pdf

[Figure]

**Supplement:**

**General response**

**Comment**
**1**. My main criticism of the paper is that the model used has not been adequately explained. It would not be possible for someone using the same or a different model to reproduce the work without a great deal of extra information. In my specific comments, I try to identify and ask questions at points in the text where vague language is used, or where important details are omitted. To allow comparison of their work, I encourage the authors to be as explicit and detailed as possible about how they achieved their results. Examples of open questions regarding how the model works:

**Line70:** what is meant by "double-layer" explicit solution? Is the Qfrost software publically available (e.g. on GitHub) – if so, why not reference it? If the model is well-documented, this might be a sufficient means of answering many of the questions regarding the method.

**Response**
Most likely, the comment appeared due to incorrect translation, since the explicit two-layer scheme, the method of balances and the enthalpy formulation of Stefan's problem are all standard terms used for numerical calculations of the ground heat transfer. Therefore, «the explicit two-layer scheme» has to be used instead of a « double-layer explicit solution». So the corrected sentence would be: «The explicit two-layer scheme is applied using the balance method and the enthalpy formulation of the problem».

Unfortunately, the Qfrost software for geocryological modeling (Certificate of the State Registration No. 2016614404 of 22) is no more available on-line despite the fact that according to the idea of its main developer Denis Pesotsky it had to be publicly available for free (previously, the program was posted at http://www.qfrost.net).  But due to his early death, the web page no longer works and the software and its usage and distribution are limited to his colleagues and collaborators at Moscow University. Nevertheless, we hope to make it available soon.  But the code is in open access on GitHub  https://github.com/kriolog/qfrost . We are not sure it is worth inserting into the article since there is a large part of the information in Russian
* * *
**Comment**
**Line 80**: "extrapolated" – what method was used to extrapolate results from monolithic stratigraphies to more complex stratigraphies? Introducing changes in porosity, grain size, thermal properties, etc. would presumably change the temperature field solutions?

**Response**

We consider two cases to extrapolate our results:
In the first case, we consider the uniform alternation of homogeneous layers with a relatively low thickness (relative to the total thickness of the permafrost).  The linear interpolation is simply performed in accordance with the percentage of the thickness of permafrost obtained during modeling for two "pure" soils at a given moment.

The second case relates to the two-layer structure of the section when a sufficiently thick (as compared with the permafrost thickness) homogeneous layer is underlined by a second homogeneous layer of unlimited thickness. Then the following considerations are valid. If the thickness of the upper layer is zero, then the thickness of the permafrost is equal to the result of simulation for "pure" rocks/sediments of the second layer. If the thickness of the upper layer is equal to (or more) the thickness of the permafrost obtained for the first layer, then the problem

becomes single-layer and the thickness of the permafrost is equal to that of the first layer. Therefore, when the thickness of the upper layer changes from zero to the thickness of the permafrost of the first layer, the thickness of the two-layer section changes from the thickness of the permafrost of the second layer to the permafrost thickness the first layer. Therefore, as a first approximation, this change is linear (which is not entirely true) and a simple linear interpolation formula can be obtained.
* * *
**Comment**

On **line 82**, all rocks (please replace) were assumed to be saline – why? how saline was the sediment assumed to be? Did salinities vary with depth? Was salt diffusion permitted? How did salt content affect the freezing characteristic curve or liquid water of frozen material? - How are discontinuities avoided at the borders between domains/subdomains/areas/subareas?

**Response**
A very high degree of averaging over the properties was used for the modeling caused by the lack of data on the water area. The rare drilling data showed the salinization of sediments through the entire drilling depth. So there was not possible to take into account the salt diffusion, and the salinity did not vary with the depth

We will add the next text after the sentence « All rocks and sediments were assumed to be saline from top to bottom of the modeling domain» Line 82 :

All reference rocks and sediments were considered saline; the concentration of pore saline solution corresponding to that of seawater (23 ‰) was used. Thermophysical properties, the content of unfrozen water and the heat of the phase transitions of water in the pores, the freezing point of the deposits was set taking into account the indicated salinization.

As for the question about the borders between domains/subdomains/areas/subareas we suggest to insert the following text in the section that describes the methodology:

The zoning is determined by the allocation of territorial units that are characterized by uniformity of formation conditions and, accordingly, by similar (on the given scale of the studies) parameters of permafrost: its distribution, thickness, depths of the top (Kudryavtsev V.A.( Ed.) Methods of Cryogenic Survey. Moscow State University, Moscow, 358 pp, 1979 (in Russian).
* * *
**Comment**
As a result, claims are made in the paper, but there is not enough information given to the reader to be able to judge whether the claim is justified or on what basis it has been made. For example: - section 4.2 lists 8 controls on the "pattern of permafrost distribution", but 2 of the 8 (lithology and properties or rocks, and Holocene climate optimum) are not described in any detail, making it impossible for the reader to follow the argument or design studies that reproduce the work. A 3rd control (thermal effect of river waters) is not even modeled, so it is not clear how the authors can conclude that this acts as a control. It seems to be an assumption in the model design, but not enough information is provided for the reader to be able to judge. -

**Response**

Thank you, you are absolutely right, the influence of control factors within the periglacial is not visualized. Therefore, we are going to give four figures showing the dynamics of the Kara shelf permafrost (top and base) for the past 125 Kyr. The results obtained during modeling for the following aspects for the southwestern and northeastern areas will be presented: 1) different deposit temperatures within the same sea depths, lithology, heat flux etc. (eg. in cryochrons (for example, MIS-2) in the NE it's 11 degrees lower than in the SW) 2) the same for the different sea

depths (120 and 5 m), 3) different heat flux, 4) different lithology ( eg. sand and clay silt). We did not model the influence of the dammed basin, but it is shown in Fig.7 and described in the paper above Figure 7. The influence of rivers is described according to actual data. An increase in water temperature in the Holocene optimum took place only in the southwestern part, the heated water came from the Barents Sea and is also described in the paper.
* * *
**Comment**
Tables 2-7 lists the model output for "depths to permafrost top" – but what is meant by "permafrost top" has not been defined anywhere. Does this correspond to an isotherm, the presence of any ice, or of some minimum amount of ice? Or does the model output the depth of the phase change boundary?
**Response**
The permafrost top corresponds to the -1.8 isotherm, according to the saline concentration in the seawater that was set for the pores of the deposits. This would be added to the text.
* * *
**Comment**
3. The language used in the paper is sometimes imprecise or even incorrect; the paper should be proof-read by an native-speaker with some background in the topic. As examples: a. "cold": is probably being used to refer to cryotic conditions, or to conditions below the freezing point. As it stands, it is a vague descriptor. b. "rock": is used to refer to earth material, including either rock or sediment, consolidated or unconsolidated material. In English, "rock" is used to refer only to bedrock material, and would exclude sedimentary deposits of terrestrial, marine or other origin. As it stands, all instances of the use of "rock" need to be replaced with something more precise. c. More examples are given below in the specific comments.

**Response**
Accepted, thank you for a very important comment: «cold» will be changed to «cryotic»:

As for the rocks, we would replace the «rock» for the «ground» (i.e. soil and rock) or «deposits» wherever it is necessary
* * *
**Comment**
4. The abstract is extremely short and does not provide enough information for a reader
to decide whether he/she wants to read the paper. It needs to introduce the larger context for the study, the central question/focus/hypothesis, more detail on the method. It should report key results, findings and conclusions, and may suggest implications or outlook based on the study.

**Response**
Accepted, the abstract is extended according to the comment:

The evolution of permafrost in the Kara shelf is reconstructed for the past 125 Ka. The work includes zoning of the shelf according to geological history, compiling sea-level and ground temperature scenarios within the distinguished zones, and forward modeling to evaluate the thickness of permafrost and the extent of frozen, cryotic and unfrozen ground. Special attention is given to the scenarios of the evolution of ground temperature in key stages of history that determined the current state of the Kara shelf permafrost zone: characterization of the extensiveness and duration of the existence of the sea during the marine isotope stage MIS -3, the spread of glaciation and dammed basins in MIS-2. The present shelf is divided into continuous,

discontinuous to sporadic, and sporadic permafrost. Cryotic deposits occur at the west and northwest water zone and correspond to areas of MIS-2 glaciation. Permafrost occurs in the periglacial domain that is a zone of modern sea depth from 0 to 100 m, adjacent to the continent. The distribution of permafrost is mostly sporadic in the southeast of this zone, while it is mostly continuous in the northeast. The thickness of permafrost does not exceed 100 m in the southeast and ranges from 100 to 300 m in the northeast. Unfrozen deposits are confined to the estuaries of large rivers and the deepwater part of the St. Anna trench. The modeling results are correlated to the available field data and are presented as geocryological maps. The formation of frozen, cryotic and unfrozen ground of the region is inferred to depend on the spread of ice sheets, sea level, and duration of shelf freezing and thawing periods
* * *
**Comment**
5. The reference list is incomplete. Vasiliev & Rekant (2018) are missing, for example. The reference list needs to be cross-checked with the submitted paper. Some reference citations still include initials (see Fig. 4 caption, for example).

**Response**
Accepted, thank you

Line 32 «Vasiliev & Rekant (2018)» changed to «Vasilyev et al., 2018»
Figure caption to Fig. 2 was corrected:
«Baranskaya and Romanenko et al. (2018)» changed to «Baranskaya et al. (2018)»
Gusev et al. (2013a) changed to Gusev et al. (2013) and the according reference was corrected:
Gusev E.A., Bolshiyanov D.Yu., Dymov V.A., Sharin V.V., Arslanov Kh.A.: Holocene marine terraces of southern islands of the Franz Joseph Land archipelago. Problemy Arktiki i Antarktiki, 97 (3), 103-108, 2013
the next references were added to the reference list:
Gusev E. A., Molod'kov A. N. Structure of sediments of the final stage of the Kazantsevo transgression (MIS 5) in the north of Western Siberia. Doklady Earth Sciences, 443(2), 458-461, 2012
Romanovskii, N.N., Hubberten, H.-W., Gavrilov, A.V., Eliseeva, A.A., Tipenko, G.S. Offshore permafrost and gas hydrate stability zone on the shelf of East Siberian Seas // Geo-Marine Letters, 2005, v. 25, N 2-3, p. 167-182.
Kudryavtsev V.A.( Ed.) Methods of Cryogenic Survey. Moscow State University, Moscow,358 pp, 1979 (in Russian).

the next references were deleted:
Gusev E.A., Anikina N.Yu., Arslanov Kh.A., Bondarenko S.A., Derevyanko L.G., Molod'kov A.N., Pushina Z.V., RekantP.V., Stepanova G.V.: Quaternary tratigraphy and pleogeography in Sibiryakov Island for the past 50 000 years. Proceedings of the Russian Geographical Society, 145 (4), 65-79, 2013b.
Gusev, E.A., Sharin, V.V., Dymov, V.A.,Kachurina, N.V., Arslanov, Kh.A, 2012. Shallow sediments in the northern Kara shelf: New data. Razvedka i Okhrana Nedr, 8, 87-90, 2009.
Trofimov V.T. (Ed.): The Yamal Peninsula (Engineering-Geological Review). Moscow State University, Moscow, 278 pp,1975 (in Russian).
The reference «Flint R.F.: Glacial and Pleistocene geology. N.Y., J. Wiley a. sons, 553 pp., 1957» was moved from Line 485 to 473
Lines 585-589 the reference was corrected : Streletskaya I.D., Shpolanskaya N.A., Kritsuk L.N., Surkov A.V.: Cenozoic deposits in the Yamal Peninsula and the problem of their origin. Vestnik Moscow University, Ser. 5. Geografiya 3, 50-57, 2009

Lines 620-624  Yershov E.D. changed to  Yershov E.D. (Ed.)
The initials from Fig. 4 caption were deleted
The article has been carefully checked and corrected in accordance with the changed links.
* * *
**Comment**
6. This paper stays true to the general phenonmenon of Russian authors citing mostly Russian work, and Canadian/Alaskan researchers citing mostly North American. For citations dealing with regionally specific processes, this is understandable. But neglecting to look at how the North American community has approached modelling the exact same processes under different conditions is harmful in two ways: it exposes the work to the criticism of being too narrow in its approach, and it makes it less likely that the work will be found and cited by North Americans. I encourage the authors to show their familiarity with the field by referring to the work of researchers from outside of their region, who have presented novel ideas in the field of subsea permafrost modelling. Some examples: - Whitehouse, P. L., Allen, M. B., Milne, G. A. Glacial isostatic adjustment as a control on coastal processes: an example from the Siberian Arctic, Geology, 35,747–750,doi:10.1130/G23437A.1,2007. –anything from the group of Romanovsky and Nicholsky (e.g. Nicolsky, D., Shakhova, N. Modeling sub-sea permafrost in the East Siberian Arctic Shelf: the Dmitry Laptev Strait. Environmental Research Letters, 5(1), 15006, 2010.). - anything from Taylor, A. (e.g. Taylor, A. E., S. R. Dallimore, P. R. Hill, D. R. Issler, S. Blasco, Wright, F. Numerical model of the geothermal regime on the Beaufort Shelf, Arctic Canada since the Last Interglacial, J. Geophys. Res. Earth Surf. , 118, doi:10.1002/2013JF002859, 2013.).

1) This phenomenon on mathematical modeling is quite understandable. Soviet (Russian) permafrost scientists started the modeling back in the 1960s. (Sharbatyan A.A. To the history of the development of permafrost (on the example of the West Siberian Plain) // Transactions of the Institute of permafrost. Academy of Sciences of the USSR, v. XIX, 1962). Modeling was carried out on extremely slow analog devices. For the shelf, the first modeling was carried out in 1969 (Molochushkin E.N. Thermal conditions of rocks in the southeastern part of the Laptev Sea. Abstract of thesis , 1970). The methodology and simulation results were widely debated in the former USSR in the late 1970s – early 80s. Therefore, the literature on modeling in Russian is at least an order of magnitude more extensive than in English, which, in fact, determines its greater citation.

However, it should be noted that Russian-language literature is widely represented in the international database. In addition to the references cited in this article, the state of mapping of the submarine cryolithozone by Russian researchers at the turn of the 20th and19th centuries is characterized in the next paper:  Gavrilov A.V. (2001) Geocryological Mapping of Arctic Shelves. In: Paepe R., Melnikov V.P., Van Overloop E., Gorokhov V.D. (eds) Permafrost Response on Economic Development, Environmental Security and Natural Resources. NATO Science Series (Series 2. Environment Security), vol 76. Springer, Dordrecht.

In current work the authors used the developments from N.E. Shakhova and D. Nikolsky for construction the scenarios. Moreover, D. Nicolsky et al., 2012 used data of modern warming to set the Holocene temperature of bottom water,          while we did set more realistic data - the reconstructed temperatures in each of the Holocene warmings and coolings. Besides, in the present paper, the authors tried to take into account the increase in the temperature of bottom water in the coastal zone warmed up in summer, which occurred during the transgression of the sea on each section of the shelf
* * *
**Specific comments:**
**Comment**
Line 13: the use of as a unit does not follow SI.
**Response**
Kyr corrected to Ka
* * *
**Comment**
Line 24: "In the latest ... earliest ..." needs correction.
**Response**
By late 1970s - early 1980s
* * *
**Comment**
Line 45: "raised high" – please quantify
**Response**
Corrected, +45, +55,+60 m
* * *
**Comment**
Line 49: add "and" and remove "and so on"
**Response**
Done
* * *
**Comment**
Line 58: replace "provide their progress" with "extend their work"?
**Response**
Thank you, we changed this according to your suggestion.
* * *
**Comment**
Line 59: what is meant with "geocryological results"? Please specify
+
Line 61: "obtained estimates" – of what? Please specify
**Response**
corrected sentence:
The work includes compiling a database of paleogeographic, geological, tectonic, and geocryological conditions used further for dividing the region according to geological history and for creating possible scenarios of sea level and ground temperature variations that serve as boundary conditions in heat transfer modeling
* * *
**Comment**
Fig. 1. This figure provides an overview of the method, but uses many general or non-specific terms that reduce the amount of information communicated: - in the top box, what is meant by "environmental data"? - in the second box, what is meant by "conditions"? - in the left third box, delete "dynamics" (adds nothing to "history") - in the third right box and in the fourth left box, replace"rocks"-in the fourth right box: "density of the heat flow from the depths" is usually referred to "geothermal heat flux" - in the fifth box: "Testing ... of the model" is almost entirely free of content. How was which model tested? More specific word choice could make this box informative - the lowest box is actually two steps: "coordination" and "mapping" - what is meant by "coordination"? This question is never really answered in the paper, but is critical for understanding what was done. Does the model output get changed in some way by comparison with field data? Where and where not? How? These are important points for anyone wanting to reproduce or apply the method in the same or in other geographical regions.
**Response**
The figure will be adjusted in accordance with the comments using more specific terms

**Comment**

Line 72: "Permafrost dynamics were..."

**Response**

Sorry, but we don't get why it has to be plural here

**Comment**

Line 73: "including..." suggests that other scenarios/conditions were NOT included? How many and why not?

**Response**

Thank you for the notice. The sentence will be corrected to sound more clear:

The permafrost dynamics was simulated for numerous paleoclimate scenarios that cover the full range of presumable conditions in the Arctic shelves. The total number of paleo-scenarios for the Kara sea used in the course of mathematical modeling was 40.

**Comment**

Line 77: how were regions of different geothermal heat flux mapped or determined?

**Response**

In accordance with the data of geothermal studies (Khutorskoy et al., 2013), the Kara shelf is characterized by a heat flux density of 50 to 75 mW /m$^2$. There is very few no point-referenced data. Therefore, the technique involves modeling for two extreme density values. Based on the simulation results and actual data, it was possible to draw conclusions about the heat flux values characteristic of various tectonic structures.

**Comment**

Line 79 and in all following text: modelling was probably not restricted to the "rock".

**Response**

corrected

The modeling was performed for several uniform reference rock and soil types in order to reduce the number of possible solutions in the conditions of high lithological diversity in the area.

**Comment**

Line 87: "subsea permafrost had presumably fully degraded..." – this statement requires a reference, especially in light of modelling, for example by Romanovsky, N. N., showing permafrost elsewhere persisting through interglacials; this point is important, since other researchers have shown that a systematic bias in model results is obtained depending on the initial conditions. Such results show that setting permafrost to zero at the interglacial will introduce a warm bias, that at least would need to be tested.

**Response**

Romanovsky's conclusions are about the Laptev Sea shelf, where there are almost no marine terraces of MIS-5e. The very different conditions occured for the Kara region, where the entire north of the West Siberian Plain had been covered by the sea from 140 to 120 Ka.

Link to MIS-5e : The State Geological Map of the Russian Federation Scale 1:1000000 (third generation). Ser. West Siberia, Sheet R-42, Yamal Peninsula, 2015 (in Russian).; Map of Quaternary deposits in the Russian Federation, scale 1:2 500 000, Explanatory note. Minprirody, Rosnedra, VSEGEI, VNII Okeangeologiya, 2010 (in Russian).

**Comment**

Fig. 2: it looks like only 14 sites out of more than 100 are located on the shelf, i.e. pertain to subsea permafrost. Is this correct? Please add a description of the red line (which is currently not described until Fig. 8).

**Response**

Yes, it is correct. There are few publications containing data on submarine permafrost in Fig. 2. They are actually few. But the figure is called: Late Pleistocene geology of the Kara region: data coverage. They are the paleogeographic data necessary for scenario and mathematical modeling. The red line description, which is   research region boundary will be added on Fig 2,3,6

───────────────────────────────

**Comment**
Line 106: "the existence of a number of idea about its development..." is not a peculiarity of any region, it is true of every region!
**Response**
It's true, thank you. Corrected sentence:
There is number of ideas about the paleogeography of the Kara region  and its development in the Late Pleistocene.

───────────────────────────────

**Comment**
Line 115: dammed lakes are invoked to explain the unfrozen zone. Why is the sensible and convective heat transport at the river bed and in the estuarine regions not sufficient to explain the absence of frozen material? Surely the rivers maintain and have maintained positive benthic temperatures for long periods?
**Response**
The riverbeds are lines. Here there is a large area of the flatten bottom and river valleys are traced incomparably worse than on the periglacial shelf of the Laptev and East Siberian seas. There bathymetry shows that the rivers functioned the entire period of MIS-2. But it is absent on the West Siberian shelf. Instead of river deltas to the west and east of the coast of Western Siberia, the estuaries of the Ob, Taz, and other rivers deeply protrude into the land. And even the small river Gyda has an estuary, the width of which is almost the same as the length of the Gydy river, and the length of the estuary is like 3-4 Gyda rivers. This has to be an ice-barrier basin. This is an assumption. But it is well confirmed by talik on the water continuation of the rivers of Western Siberia, and estuaries deeply protruding into the land.  The question is how could the Ob and the Yenisei, flowing through the whole Western and Middle Siberia, carrying a mass of sediment (since they  drain the Altai and Sayan mountain systems), form not a delta, such as Lena, but estuaries? In our opinion, the assumption is very realistic.

───────────────────────────────

**Comment**
Line 119: "Insignificance of the severity" is convoluted language that should be simplified.
**Response**
Accepted, changed to «…the poor expression of the ancient valley network…»

───────────────────────────────

**Comment**
Line 154: explain the abbreviation "MMP"
**Response**
corrected. It's permafrost now

───────────────────────────────

**Comment**
Line 156: "sea level" rather than "sealevel"
**Response**
corrected

───────────────────────────────

**Comment**
Line 201: on what basis was it decided how long each portion of the shelf spent in the coastal zone (400-2000 years)? Why were waters in this zone saline? – is this not the zone most affected by the freshwater layer above the halocline, by snow melt and river runoff? Dmitrenko et al (2011) show

the freshwater nature of the coastal zone in the Laptev Sea. And why was this zone warmer? Bedfast ice can result in cooling of the seabed from 0 – 2 m water depth. A little more justification and specification of these boundary conditions, which determine the most immediate and rapid response of permafrost to inundation by seawater, are necessary.

**Response**

It is known that at sea depths from 2 to 7 m in the 1970s (Zhigarev, 1981), up to 10 m in the 2000s. (Dmitrenko) the mean annual temperature of bottom waters in the Laptev Sea is positive and bottom sediments thaw from above. For the Kara Sea, there are no such data on isobath intervals, but it is known that temperatures are generally lower. Therefore, we assumed that the interval of isobaths with such water temperatures was limited to 5-6 m. Episodes with such water temperatures occurred on the shelf during the postglacial transgression even in its initial periods, since the July temperature is reconstructed for pre-Holocene warming (allered) exceeding 2 ° C the temperature of the 1980s. (Velichko et al., 2000). Therefore, we constructed scenarios for these episodes. To determine their duration, dated data on the absolute altitudes of the sea level during the transgression of the Laptev Sea were used (Bauch et al., 2001). The transgression rate was not the same. The shortest episodes (400 and 375 years) of positive temperatures are determined for periods 15-11 and 10-9 ka respectively. The longest intervals were 11–10, 9–5, and 5–0 Ka with 1000, 750 and >5000 years according to simple calculations.
* * *
**Comment**

Fig. 8: The map shows permafrost thickness, which can clearly result as output from 1D numerical modelling. What conditions were applied to determine zonation of permafrost based on continuity (continuous,discontinuous,sporadic)? I.e. how do conclusions about distribution result from 1D modelling? Caption: why are only "fragments" of the map shown? Why not present the reader with the whole map?

**Response**

A paleogeographic scenario was constructed, consisting of a series of paleotemperature curves (or scenarios), in which the main factors that formed the modern permafrost were taken into account. Among these factors were: zoning, sectoral, fluctuations in sea level and its depth, which determined the period of freezing during shelf drainage and the thawing period when it was in the flooded state, the period of glaciation and thickness of the ice sheet, geothermal gradient, ground composition and properties, area, within which the water temperature rose at the Holocene optimum etc. In accordance with the indicated curves, modeling was carried out, the results of which reflected the influence on the permafrost of all of the above factors.

The whole map is being prepared for publication in the atlas

---

## Author Comment (AC2) · 22 Jan 2020

**Comment**

1. Plot the air temperature (ground surface temperature) for different regions for the last 125k years. As I understand there is a temperature zonation factor involved. What is it? How does the air temperature differ between various regions, subzones?

**Response**

The article presents graphs of mean annual ground temperatures over the past 125 thousand years depending on paleogeographic events (shelf drying, flooding, glaciation, etc.)according to latitudinal zoning and meridional sectorality (Fig. 4,5,7). When specifying latitudinal zonality during periods of shelf drying, the authors followed differences in ground temperatures reflected on the Russian Geocryological Map (Yershov ed., 1991). The mean annual ground temperature is 4-6 °C lower in the NE of the region (north of Taimyr) adjacent to the Kara coast, than as for the SW (south-west of Yamal). When zoning the shelf (Table 1, Fig. 3), the southwest (68-71 ° N) and north-eastern (72-77 ° N) areas were distinguished. During the periods of drainage in the periglacial part of the shelf in MIS-2, the following values for the ground temperature were taken: -19 ° C for the north-eastern area, and -15 °C for the south-western one .

Reference:

Yershov E.D. (Ed.): Geocryological Map of the USSR, scale 1:2 500 000. Moscow State University, Moscow, 1991 (in Russian).
* * *
**Comment**

2. It is stated that "about 50-75ky bp the sea level was the same as present". Unfortunately, the paper lacks the sea level dynamics,I think it was used in the scenario building, but not explicitly shown. Please supplement Figure 4 with a plot of the relative sea level curve with respect to the present-day conditions for the last 125k years.

**Response**

The paper claims that the sea level was the same as current 50-25 Ka BP (not 50-75, Lines 207-208). This is shown at Fig. 6 based on the marine sediments dating (50-25 Ka BP) on the islands, shelf and the sea coast . We will add the Figure showing fluctuations of the sea level during the modeling period of 125-15 Ka BP.
* * *
**Comment**

3. Make a series of maps to show which areas were flooded, glaciated or exposed to air 10, 20, 40, 55, 70, 80 ky bp. It will help a reader. It is okay to take different times, e.g. middle of the MIS periods. The goal is to help future researchers to understand when a certain part of the shelf was exposed to the air. Right now, it is not clear. Also, change the y-labels in Figure 4 to "Sea level with respect to the present-day datum, m". Make values in Figure 4a negative. '

**Response**

The authors totally agree with the reviewer: the relevant illustrative material will help the reader to understand the content of the article. Therefore, the change in surface conditions will be illustrated by a series of cartographic schemes for the following moments: MIS- 5e (130-120 ka BP); MIS -5d (117-110 ka BP); MIS- 5c (110-105 Ka BP); MIS -5b (100-75 Ka BP).

As for the Figure 4, we will change y-label for «Sea level with respect to the present-day, m », but it seems more convenient not to make changes in values in Figure 4a, because they are depth's values, not altitude as in Figure 4b.
* * *
**Comment**

4. How are the effects of salts taken into the account? Does the salt lower the freezing point depression? Do you take into the account unfrozen liquid pore water while freezing the saline water? How are the salt effects parameterized in the model? Explain and clarify in the paper.

**Response**

The freezing temperature equal to the freezing temperature of sea water ( - 1,8°C) was set for all types of soils for the modeling. This assumption was made due to the fact that all the marine sediments composing the Yamal Peninsula have the close values of the salinity to a depth of 300 m and more (Chuvilin et al., 2007).Very high degree of averaging over the properties was used for the modeling caused by the lack of data on the water area. The rare drilling data showed the salinization of sediments through the entire drilling depth. So there was not possible to take into account the salt diffusion, and the salinity did not vary with the depth. Because the modeling has evaluative nature a scheme with complete freezing (thawing) of moisture in the ground at the moving front of phase transitions was used. The content of unfrozen water in the sediments was taken into account by reducing the volumetric heat of phase

transitions in the model by the value corresponding to the average content of unfrozen water in different types of rocks at negative temperatures typical of the process under study.

We will add the next text after the sentence « All rocks and sediments were assumed to be saline from top to bottom of the modeling domain» Line 82 : «All reference rocks and sediments were considered saline; the concentration of pore saline solution corresponding to that of sea water (23 ‰) was used. The freezing temperature equal to the freezing temperature of sea water (-1,8°C) was set for all types of soils for the modeling. Thermophysical properties, the content of unfrozen water and the heat of the phase transitions of water in the pores, the freezing point of the deposits were set taking into account the indicated salinization.

Reference:

Chuvilin E.M., Perlova E.V., Baranov Yu.B., Kondakov V.V., Osokin A.B., Yakushev V.S. The structure and properties of cryolithozone sediments of the southern part of the Bovanenkovo gas condensate field. M.:"Geos", 2007, p.20
* * *
**Comment**

**5.** The paper goes into the discussion of various ground layers (sand, clay,etc) and heat flux values. I suggest moving this analysis into a separate section, e.g."Sensitivity analysis". Do you do sensitivity analysis with respect to salt concentration?

**Response**

Actually we have a related comment from the first reviewer about lack of information on the influence of control factors of permafrost dynamics ( section 4.2). Therefore, it seems to be more convenient to expand that section with four figures showing the dynamics of the Kara shelf permafrost (top and base) for the past 125 Kyr. The results obtained during modeling for the following aspects for the southwestern and northeastern areas will be presented: 1) different deposit temperatures within the same sea depths, lithology, heat flux etc. (eg. in cryochrons (for example, MIS-2) in the NE it's 11 degrees lower than in the SW) 2) the same for the different sea depths (120 and 5 m), 3) different heat flux, 4) different lithology ( eg. sand and clay silt).

As for the salinization, the modeling was provided for the uniform salt concentrations as we described in the response to the previous comment.
* * *
**Comment**

6. The manuscript is understandable, but terminology is used unconventionally

**Response**

Thank you, we will moderate the language focusing on precision and word choice. The paper would be proof-readed by a native-speaker with background in the topic.

Some obvious changes: «cold» will be changed to «cryotic»; we would replace the «rock» for the «ground» (i.e. soil and rock) or «deposits» wherever it is necessary.

---

## Author Response (AR1)

We would like to thank the reviewers and the Editor for their careful review of our manuscript. Our point-by-point responses to all the comments and annotations are presented below. The comments from the reviewers are shown in black, and our responses in red. Where indicated, additions to the manuscript are shown in blue and deleted text is indicated by red strikethrough, and line numbers are based on our revised (non-track) manuscript. Besides, we added some information about changes in the paper that are not mentioned in the response to the Reviewers They are located below the responses to the comments above the marked-up manuscript version.

1. We cropped the Figures 2, 3, 6 and 9 (previous № 8): in the previous version of the article, the map covered part of the Laptev Sea, in this version there is just the Kara Sea.
2. Fig.2: The site №12 was missed; we added it to the current version.
3. The numbers of the figures were changed caused by the additional figure.
4. We made several changes in the Reference list.

==============================================================================
Referee #1

**Comments from the reviewer #1 and our responses:**

**General response**

**Comment**
**1**. My main criticism of the paper is that the model used has not been adequately explained. It would not be possible for someone using the same or a different model to reproduce the work without a great deal of extra information. In my specific comments, I try to identify and ask questions at points in the text where vague language is used, or where important details are omitted. To allow comparison of their work, I encourage the authors to be as explicit and detailed as possible about how they achieved their results. Examples of open questions regarding how the model works:

**Line70:** what is meant by "double-layer" explicit solution? Is the Qfrost software publically available (e.g. on GitHub) – if so, why not reference it? If the model is well-documented, this might be a sufficient means of answering many of the questions regarding the method.

**Response**
→ Most likely, the comment appeared due to incorrect translation, since the explicit two-layer scheme, the method of balances and the enthalpy formulation of Stefan's problem are all standard terms used for numerical calculations of the ground heat transfer. Therefore, «the explicit two-layer scheme» has to be used instead of « double-layer explicit solution».
Unfortunately, the Qfrost software for geocryological modeling (Certificate of the State Registration No. 2016614404 of 22) is no more available on-line despite the fact that, according to the idea of its main developer Denis Pesotsky it had to be publicly available for free (previously, the program was posted at http://www.qfrost.net). But due to his early death the web page no longer works and the software and its usage and distribution is limited to his colleagues and collaborators at Moscow University. Nevertheless, we hope to make it available soon. But the code is in open access on GitHub https://github.com/kriolog/qfrost . We are not sure it is worth inserting into the article, since there is a large part of the information in Russian.

The next sentences were added (Lines 78-79):

→  The explicit two-layer scheme is applied using the balance method and the enthalpy formulation of the problem. The code used is publicly available under https://github.com/kriolog/qfrost (last access: 15 February 2020) (kriolog, 2020).

→ Reference (Lines 493-494):
→ kriolog: Software for a numerical geotechnical modeling of thermophysical processes in freezing and frozen soils, available at: https://github.com/kriolog/qfrost, last access: 15 February 2020

———————————————————

**Comment**

**Line 80**: "extrapolated" – what method was used to extrapolate results from monolithic stratigraphies to more complex stratigraphies? Introducing changes in porosity, grain size, thermal properties, etc. would presumably change the temperature field solutions?

**Response**

→ We added the results extrapolation algorithm (Lines 90-103):
→ We consider two cases to extrapolate our results:In the first case, we consider the uniform alternation of homogeneous layers with a relatively low thickness (relative to the total thickness of the permafrost). The linear interpolation is simply performed in accordance with the percentage of the thickness of frozen ground obtained during modeling for two "pure" soils at a given moment.The second case relates to the two-layer structure of the section, when a sufficiently thick (as compared with the permafrost thickness) homogeneous layer is underlained by a second homogeneous layer of unlimited thickness. Then the following considerations are valid. If the thickness of the upper layer is zero, then the thickness of the permafrost is equal to the result of simulation for "pure" rocks/sediments of the second layer. If the thickness of the upper layer is equal to (or more) the thickness of the permafrost obtained for the first layer, then the problem becomes single-layer and the thickness of the permafrost is equal to that of the first layer. Therefore, when the thickness of the upper layer changes from zero to the thickness of the permafrost of the first layer, the thickness of the two-layer section changes from the thickness of the permafrost of the second layer to the permafrost thickness the first layer. Therefore, as a first approximation, this change is linear (which is not entirely true) and a simple linear interpolation formula can be obtained.

———————————————————

**Comment**

On **line 82**, all rocks (please replace) were assumed to be saline – why? how saline was the sediment assumed to be? Did salinities vary with depth? Was salt diffusion permitted? How did salt content affect the freezing characteristic curve or liquid water of frozen material? - How are discontinuities avoided at the borders between domains/subdomains/areas/subareas?

**Response**

→ Very high degree of averaging over the properties was used for the modeling caused by the lack of data on the water area. The rare drilling data showed the salinization of sediments through the entire drilling depth. So there was not possible to take into account the salt diffusion, and the salinity did not vary with the depth.
→ We added the next text (Lines 103-107) :
→ All reference rocks and sediments were considered saline with Ds = 0.8-1.1% according to the concentration of pore saline solution corresponding to that of Kara sea bottom

waters concentration (32-34 ‰). Therefore the freezing temperature of the pore solution is close to -1.8 °C. Thermophysical properties, the content of unfrozen water and the heat of the phase transitions of water in the pores, the freezing point of the deposits were set taking into account the indicated salinization.

→ As for the question about the borders between domains/subdomains/areas/subareas we inserted the following text in the section that describes the methodology (Lines 73-75):

→ The zoning is determined by the allocation of territorial units that are characterized by uniformity of formation conditions and, accordingly, by similar (on the given scale of the studies) parameters of permafrost: its distribution, thickness, depths of the top (Kudryavtsev, 1979).
* * *
**Comment**

As a result, claims are made in the paper, but there is not enough information given to the reader to be able to judge whether the claim is justified or on what basis it has been made. For example: - section 4.2 lists 8 controls on the "pattern of permafrost distribution", but 2 of the 8 (lithology and properties or rocks, and Holocene climate optimum) are not described in any detail, making it impossible for the reader to follow the argument or design studies that reproduce the work. A 3rd control (thermal effect of river waters) is not even modeled, so it is not clear how the authors can conclude that this acts as a control. It seems to be an assumption in the model design, but not enough information is provided for the reader to be able to judge. -

**Response**

→ Thank you, you are absolutely right, the influence of control factors within the periglacial is not visualized. Therefore, we added four figures showing the dynamics of the Kara shelf permafrost (top and base) for the past 125 Kyr. The figure is presented in the supplements showing the influence of zonation, heat flux, lithology and properties and the influence of sea depths. The link to the reference is at the Line 306 in the section « 4.2. Distribution of permafrost, its thickness and depth to the permafrost table»

:

→ This pattern has several controls (Supplementary Figure 1):

[Figure]

→ Supplementary Figure 1. The influence of various environmental factors on the dynamics of shelf permafrost over the past 125 kyr  according to the results of mathematical modeling
a) the influence of  latitudinal climatic zonation and division into sectors : southwestern (SW) and northeastern (NE) shelf parts (the loam, 50 m isobaths, q=50 mW/m2)
b) the influence of heat flux: 50 mW/m2  and 75 mW/m2 (the loam, 50 m isobaths, SW)
c) the influence of  lithology and properties of deposits : sand and loam ( 50 m isobath, q=50 mW/m2, SW )
d) the influence of sea depths (bottom isobaths): 5 and 50 m (the sand, q=50 mW/m2, SW)

→ We did not model the influence of the dammed basin, but it is shown at Fig.7 and described in the paper above the Figure 7. The influence of rivers is described according to actual data. An increase in water temperature in the Holocene optimum took place only in the southwestern part, the heated water came from the Barents Sea and is also described in the paper. Besides, we added the series of cartographic schemes illustrating , the change in surface conditions for the  125 kyr history of the Kara sea shelf (Lines 281-283)

[Figure]

MIS-5e (136-117 kyr BP)

MIS-5d (117-110 kyr BP)

MIS-5c (110-100 kyr BP)

MIS-5b (100-77 kyr BP)

MIS-5a (77-65 kyr BP)

MIS-4 (65-50 kyr BP)

MIS-3 (50-40 kyr BP)

MIS-2 (25-12 kyr BP)

→ Fig. 8. The series of cartographic schemes illustrating , the change in surface conditions in the Kara Sea shelf  for the following moments : MIS 5e (130-120 kyr,  MIS 5d (117-110 kyr)  MIS 5c (110-105 kyr); MIS 5b (100-75 kyr); 5a (75-65 kyr); MIS -4 (kyr); MIS -3 (50-25kyr); MIS -2 (25-15 kyr).
* * *
**Comment**

Tables 2-7 lists the model output for "depths to permafrost top" – but what is meant by "permafrost top" has not been defined anywhere. Does this correspond to an isotherm, the presence of any ice, or of some minimum amount of ice? Or does the model output the depth of the phase change boundary?

**Response**

→ This is a very right comment. The permafrost table corresponds to the -1.8° C isotherm, according to the saline concentration in the seawater that was set for the pores of the deposits. Now it is shown in the abstract describing the salinity as mentioned above: (Lines 103-107) :

→ All reference rocks and sediments were considered saline with Ds = 0.8-1.1% according to the concentration of pore saline solution corresponding to that of Kara sea bottom waters concentration (32-34 ‰). Therefore the freezing temperature of the pore solution is close to -1.8 °C. Thermophysical properties, the content of unfrozen water and the heat of the phase transitions of water in the pores, the freezing point of the deposits were set taking into account the indicated salinization.

→ as well as in the 4.1 section (Lines 289-291):

→ 4.1. Distribution of frozen, cooed, and  thawed deposits
Here we consider the permafrost table (frozen permafrost) corresponds to the -1.8° C isotherm, Therefore further in the text we mean frozen permafrost when talking about «permafrost» and the cryotic unfrozen deposits are «cooled» deposits (marine cryopeg).
* * *
**Comment**

3. The language used in the paper is sometimes imprecise or even incorrect; the paper should be proof-read by an native-speaker with some background in the topic. As examples: a. "cold": is probably being used to refer to cryotic conditions, or to conditions below the freezing point. As it stands, it is a vague descriptor. b. "rock": is used to refer to earth material, including either rock or sediment, consolidated or unconsolidated material. In English, "rock" is used to refer only to bedrock material, and would exclude sedimentary deposits of terrestrial, marine or other origin. As it stands, all instances of the use of "rock" need to be replaced with something more precise. c. More examples are given below in the specific comments.

**Response**

→ Accepted, thank you for a very important comment
«cold» is changed to «cooled» that means the cryotic unfrozen deposits known as marine cryopeg according to the Multi-Language Glossary of Permafrost (Everdingen, Robert van, ed. 1998 revised January 2002. Multi-language glossary of permafrost and related ground-ice terms. Boulder, CO: National Snow and Ice Data Center/World Data Center for Glaciology). We still have chosen «cooled», because in Russian literature the cryopeg always means an isolated cryopeg, and this can create a lot of confusion for non-native speakers. Therefore we decided to add that description «the cryotic unfrozen deposits are «cooled» deposits (marine cryopeg)».See Lines 290-291:

→ the cryotic unfrozen deposits are «cooled» deposits (marine cryopeg).

→ As for the rocks, we replaced the «rock» for the «deposits» (instead of rock and sediments) or «ground» ( ground temperature instead of rock temperature) wherever it was necessary
* * *
**Comment**

4. The abstract is extremely short and does not provide enough information for a reader to decide whether he/she wants to read the paper. It needs to introduce the larger context for the study, the central question/focus/hypothesis, more detail on the method. It should report key results, findings and conclusions, and may suggest implications or outlook based on the study.

**Response**

→ Accepted, the abstract is extended according to the comment (Lines 13-26):

→ Abstract. ~~The evolution of permafrost in the Kara shelf is reconstructed for the past 125 Kyr. The work includes zoning of the shelf according to geological history, compiling sea-level and ground temperature scenarios within the distinguished zones, and forward modeling to evaluate the thickness of permafrost and the extent of frozen, cold and unfrozen rocks. The modeling results are correlated to the available field data and are presented as geocryological maps. The formation of frozen, cold, and unfrozen rocks of the region is inferred to depend on the spread of ice sheets, sea level, and duration of shelf freezing and thawing periods.~~ The evolution of permafrost in the Kara shelf is reconstructed for the past 125 kyr. The work includes zoning of the shelf according to geological history, compiling sea level and ground temperature scenarios within the distinguished zones, and modeling to evaluate the thickness of permafrost and the distribution of frozen, cooled and -thawed deposits. Special attention is given to the scenarios of the evolution of ground temperature in key stages of history that determined the current state of the Kara shelf permafrost zone: characterization of the extensiveness and duration of the existence of the sea during the marine isotope stage MIS -3, the spread of glaciation and dammed basins in MIS-2. The present shelf is divided into areas of continuous, discontinuous to sporadic, and sporadic permafrost. Cooled deposits occur at west and northwest water zone and correspond to areas of MIS-2 glaciation. Permafrost occurs in the periglacial domain that is a zone of modern sea depth from 0 to 100 m, adjacent to the continent. The distribution of permafrost is mostly sporadic in the southwest of this zone, while it is mostly continuous in the northeast. The thickness of permafrost does not exceed 100 m in the southeast , and ranges from 100 to 300 m in the northeast. Thawed deposits are confined to the estuaries of large rivers and the deepwater part of the St. Anna trench. The modeling results are correlated to the available field data and are presented as geocryological maps. The formation of frozen, cooled and -thawed deposits of the region is inferred to depend on the spread of ice sheets, sea level, and duration of shelf freezing and thawing periods.
* * *
**Comment**

5. The reference list is incomplete. Vasiliev & Rekant (2018) are missing, for example. The reference list needs to be cross-checked with the submitted paper. Some reference citations still include initials (see Fig. 4 caption, for example).

**Response**

→ Accepted, thank you
The reference list and the article has been carefully checked and corrected in accordance with the changed links.
* * *
**Comment**

6. This paper stays true to the general phenonmenon of Russian authors citing mostly Russian work, and Canadian/Alaskan researchers citing mostly North American. For citations dealing with regionally specific processes, this is understandable. But neglecting to look at how the North American community has approached modelling the exact same processes under different conditions is harmful in two ways: it exposes the work to the criticism of being too narrow in its

approach, and it makes it less likely that the work will be found and cited by North Americans. I encourage the authors to show their familiarity with the field by referring to the work of researchers from outside of their region, who have presented novel ideas in the field of subsea permafrost modelling. Some examples: - Whitehouse, P. L., Allen, M. B., Milne, G. A. Glacial isostatic adjustment as a control on coastal processes: an example from the Siberian Arctic, Geology, 35,747–750,doi:10.1130/G23437A.1,2007. –anything from the group of Romanovsky and Nicholsky (e.g. Nicolsky, D., Shakhova, N. Modeling sub-sea permafrost in the East Siberian Arctic Shelf: the Dmitry Laptev Strait. Environmental Research Letters 5(1), 15006, 2010.). - anything from Taylor, A. (e.g. Taylor, A. E., S. R. Dallimore, P. R. Hill, D. R. Issler, S. Blasco, Wright, F. Numerical model of the geothermal regime on the Beaufort Shelf, Arctic Canada since the Last Interglacial, J. Geophys. Res. Earth Surf. , 118, doi:10.1002/2013JF002859, 2013.).

→ This phenomenon on mathematical modeling is quite understandable. Soviet (Russian) permafrost scientists started the modeling back in the 1960s. (Sharbatyan A.A. To the history of the development of permafrost (on the example of the West Siberian Plain) // Transactions of the Institute of permafrost. Academy of Sciences of the USSR, v. XIX, 1962). Modeling was carried out on extremely slow analog devices. For the shelf, the first modeling was carried out in 1969 (Molochushkin E.N. Thermal conditions of rocks in the southeastern part of the Laptev Sea. Abstract of thesis , 1970). The methodology and simulation results were widely debated in the former USSR in the late 1970s – early 80s. Therefore, the literature on modeling in Russian is at least an order of magnitude more extensive than in English, which, in fact, determines its greater citation.

However, it should be noted that Russian-language literature is widely represented in the international database. In addition to the references cited in this article, the state of mapping of the submarine cryolithozone by Russian researchers at the turn of the 20th and19th centuries is characterized in the next  paper:  Gavrilov A.V. (2001) Geocryological Mapping of Arctic Shelves. In: Paepe R., Melnikov V.P., Van Overloop E., Gorokhov V.D. (eds) Permafrost Response on Economic Development, Environmental Security and Natural Resources. NATO Science Series (Series 2. Environment Security), vol 76. Springer, Dordrecht.

In current work the authors used the developments from N.E. Shakhova and D. Nikolsky for construction the scenarios. Moreover, D. Nicolsky et al., 2012 used data of modern warming  to set the Holocene temperature of bottom water,        while we did set more realistic data - the reconstructed temperatures in each of the Holocene warmings and coolings. Besides, in the present paper, the authors tried to take into account the increase in the temperature of bottom water in the coastal zone warmed up in summer, which occurred during the transgression of the sea on each section of the shelf
* * *
**Specific comments:**
**Comment**
Line 13: the use of Kyr as a unit does not follow SI.
**Response**
→ «Kyr» corrected to «kyr» everywhere through the text
* * *
**Comment**

Line 24: "In the latest ... earliest ..." needs correction.

**Response**

→ Line 32 :  By late 1970s - early 1980s,
* * *
**Comment**

Line 45: "raised high" – please quantify

**Response**

→ Line 53: Corrected, «…which are raised high (+45, +55,+60 m , Bolsiyanov et al., 2006)»
* * *
**Comment**

Line 49: add "and" and remove "and so on"

**Response**

→ Done

→ Line 57 «…glacio-isostatic movements and fluctuations in sea level. »
* * *
**Comment**

Line 58: replace "provide their progress" with "extend their work"?

**Response**

→ Thank you, we changed this according to your suggestion.

→ Lines 64-65 « We follow these methods in our  research and are trying to  extend their work»
* * *
**Comment**

Line 59: what is meant with "geocryological results"? Please specify

+

Line 61: "obtained estimates" – of what? Please specify

**Response**

→ The sentences were corrected (Lines 65-68):

→ The work_ includes compiling a database of paleogeographic, geological, tectonic, and geocryological  conditions used further to divide the region according to geological history and for creating possible scenarios of sea level and ground temperature variations that serve as boundary conditions in heat transfer modeling. The general scheme of the methodology is presented in Figure 1.

**Comment**

Fig. 1. This figure provides an overview of the method, but uses many general or non-specific terms that reduce the amount of information communicated: - in the top box, what is meant by "environmental data"? - in the second box, what is meant by "conditions"? - in the left third box, delete "dynamics" (adds nothing to "history") - in the third right box and in the fourth left box, replace "rocks"-in the fourth right box: "density of the heat flow from the depths" is usually referred to "geothermal heat flux" - in the fifth box: "Testing ... of the model" is almost entirely free of content. How was which model tested? More specific word choice could make this box informative - the lowest box is actually two steps: "coordination" and "mapping" - what is meant by "coordination"? This question is never really answered in the paper, but is critical for understanding what was done. Does the model output get changed in some way by comparison

with field data? Where and where not? How? These are important points for anyone wanting to reproduce or apply the method in the same or in other geographical regions.

**Response**

→ The figure was adjusted in accordance with the comments using more specific terms (Line 69)

[Figure]

→ The testing consists in the production of mathematical modeling for an area provided with actual geocryological data. During modeling, adjustments are made to the paleotemperature scenario or/and geological model until the model data matches the actual. The adjusted scenario or/and model are used for the final simulation. This information is located within the Lines 107-108 : «The paleotemperature scenarios and the geological-tectonic model were tested by comparing the present permafrost state estimated by forward modeling with the available field data from well documented areas, to achieve the best fit.» and Lines 114-115: « The modeling results were correlated with field data and both datasets were used for the final geocryological zoning of the Kara shelf region.»
* * *
**Comment**

Line 72: "Permafrost dynamics were..."

**Response**

→ Sorry, but we don't get why it has to be plural here
* * *
**Comment**

Line 73: "including..." suggests that other scenarios/conditions were NOT included? How many and why not?

**Response**

→ Thank you for the notice. The sentence was corrected to sound more clear (Lines 82-83):

→ The permafrost dynamics was simulated for numerous paleoclimate scenarios that cover the full range of presumable conditions in the Arctic shelves., including forty scenarios for the Kara shelf. The total number of paleo-scenarios for the Kara sea used in the course of mathematical modeling was 30.
* * *
**Comment**

Line 77: how were regions of different geothermal heat flux mapped or determined?

**Response**

→ In accordance with the data of geothermal studies (Khutorskoy et al., 2013), the Kara shelf is characterized by a heat flux density of 50 to 75 mW /m$^2$. There is very few no point-referenced data. Therefore, the technique involves modeling for two extreme density values. Based on the simulation results and actual data, it was possible to draw conclusions about the heat flux values characteristic of various tectonic structures. This is written in the text( Lines 85-86):

«The heat flux was assumed to be 50 mW/m2 , which corresponds to the average for most of the shelf territory or 75 mW/m2, as in zones of relatively high heat flux in gas-bearing bottom sediments (Khutorskoy et al., 2013)».
* * *
**Comment**

Line 79 and in all following text: modelling was probably not restricted to the "rock".

**Response**

corrected. Lines 87-89:

→ The modeling was performed for several uniform reference rockrock and sediment -types in order to reduce the number of possible solutions in the conditions of high lithological diversity in the area.

→ We edited this terminology everywhere in the text. Please find responses above.
* * *
**Comment**

Line 87: "subsea permafrost had presumably fully degraded..." – this statement requires a reference, especially in light of modelling, for example by Romanovsky, N. N., showing permafrost elsewhere persisting through interglacials; this point is important, since other researchers have shown that a systematic bias in model results is obtained depending on the initial conditions. Such results show that setting permafrost to zero at the interglacial will introduce a warm bias, that at least would need to be tested.

**Response**

→ Romanovsky's conclusions are about the Laptev Sea shelf, where there are almost no marine terraces of MIS-5e. The very different conditions occured for the Kara region, where the entire north of the West Siberian Plain had been covered by the sea from 140 to 120 Ka.
Link to MIS-5e: The State Geological Map of the Russian Federation Scale 1:1000000 (third generation). Ser. West Siberia, Sheet R-42, Yamal Peninsula, 2015 (in Russian).; Map of Quaternary deposits in the Russian Federation, scale 1:2 500 000, Explanatory note. Minprirody, Rosnedra, VSEGEI, VNII Okeangeologiya, 2010 (in Russian).

→ We revised the sentence as follows(Lines 111-114):

→ At that time, subsea permafrost had presumably fully degraded over the whole studied part of the Kara Sea, as the entire north of the West Siberian Plain had been covered by

the sea from 140 to 120 Ka (Zastorozhnov et al., 2010; Shishkin et al., 2015) and the temperatures of unfrozen bottom sediments approached the steady state.
* * *
**Comment**

Fig. 2: it looks like only 14 sites out of more than 100 are located on the shelf, i.e. pertain to subsea permafrost. Is this correct? Please add a description of the red line (which is currently not described until Fig. 8).

**Response**

→ Yes, it is correct. There are few publications containing data on submarine permafrost in Fig. 2. They are actually few. But the figure is called: Late Pleistocene geology of the Kara region: data coverage. They are the paleogeographic data necessary for scenario and mathematical modeling.The red line description, which is research region boundary was added on Fig 2,3,6

→ Fig. 2. Late Pleistocene geology of the Kara region: data coverage (the red line is the research region boundary). The numbers on the map correspond to the following publications:

→ Fig. 3.  Zoning of the Kara shelf according to its geological history for 125 kyr. Abbreviations are explained in the text and in Table 1. The red line is the research region boundary

→ Fig. 6. Location of onshore and offshore sections (points) with marine sediments ($^{14}$C - figures, thousand years ago) of MIS-3 in the Kara Sea shelf, after Gusev et al. (2011; 2012a, b; 2013a, b; 2016 a, b); Baranskaya et al. (2016); Vasil`chuk et al. (1984); Molodkov et al. (1987); Bolshiyanov et al. (2009). The red line is the research region boundary
* * *
**Comment**

Line 106: "the existence of a number of idea about its development..." is not a peculiarity of any region, it is true of every region!

**Response**

→ The sentence is revised accordingly (Lines 131-132):

 There is number of ideas about the paleogeography of the Kara region and its development in the Late Pleistocene.
* * *
**Comment**

Line 115: dammed lakes are invoked to explain the unfrozen zone. Why is the sensible and convective heat transport at the river bed and in the estuarine regions not sufficient to explain the absence of frozen material? Surely the rivers maintain and have maintained positive benthic temperatures for long periods?

**Response**

→ We invoke the dammed lakes to explain the unfrozen zone based on the next assumptions and speculations: The riverbeds are lines. Here there is a large area of the flatten bottom and river valleys are traced incomparably worse than on the periglacial shelf of the Laptev and East Siberian seas. There bathymetry shows that the rivers functioned the entire period of MIS-2. But it is absent on the West Siberian shelf. Instead of river deltas to the west and east of the coast of Western Siberia, the estuaries of the Ob, Taz, and other rivers deeply protrude into the land. And even the small river Gyda has an estuary, the width of which is almost the same as the length of the Gydy river, and the length of the estuary is like 3-4 Gyda rivers. This has to be an ice-barrier basin. This is an assumption. But it is well confirmed by talik on the water continuation of the rivers of

Western Siberia, and estuaries deeply protruding into the land. The question is how could the Ob and the Yenisei, flowing through the whole Western and Middle Siberia, carrying a mass of sediment (since they drain the Altai and Sayan mountain systems), form not a delta, such as Lena, but estuaries? In our opinion, the assumption is very realistic.
* * *
**Comment**

Line 119: "Insignificance of the severity" is convoluted language that should be simplified.

**Response**

→ Accepted, changed (Lines 144-145) :

→  The poor expression of the ancient valley network is especially clearly seen when comparing it with that in the eastern sector of the Arctic, where there were no glaciations
* * *
**Comment**

Line 154: explain the abbreviation "MMP"

**Response**

→ corrected. (Lines185-186):

→ The modeling we carried out for the G-1 area showed that, to date, the  permafrost formed in the MIS-2 have not survived.
* * *
**Comment**

Line 156: "sea level" rather than "sealevel"

**Response**

→ corrected (Line 187):

→ «…the effect of climate-driven eustatic sea level change. The sea level…»
* * *
**Comment**

Line 201: on what basis was it decided how long each portion of the shelf spent in the coastal zone (400-2000 years)? Why were waters in this zone saline? – is this not the zone most affected by the freshwater layer above the halocline, by snow melt and river runoff? Dmitrenko et al (2011) show the freshwater nature of the coastal zone in the Laptev Sea. And why was this zone warmer? Bedfast ice can result in cooling of the seabed from 0 – 2 m water depth. A little more justification and specification of these boundary conditions, which determine the most immediate and rapid response of permafrost to inundation by seawater, are necessary.

**Response**

→ Thank you, we added a detailed explanation to the text (Lines 237-251)

→ During transgressions in the Holocene, each shelf part stayed for 400-2000 years in the coastal zone where bottom sediments were flooded with saline and warm near-bottom water. It is known that at sea depths from 2 to 7 m in the 1970s (Zhigarev, 1981), up to 10 m in the 2000s. (Dmitrenko) the mean annual temperature of bottom waters in the Laptev Sea stays positive and bottom sediments thaw from above. For the Kara Sea, there are no such data on isobath intervals, but it is known that temperatures are generally lower. Therefore, we assumed that the interval of isobaths with such water temperatures was limited to 5-6 m. Episodes with such water temperatures occurred on the shelf during the postglacial transgression even in its initial periods, since the July temperature is reconstructed for pre-Holocene warming (allered) exceeding  2 ° C  the temperature of the 1980s. (Velichko et al., 2000). Therefore, we constructed scenarios for these episodes. To determine their duration, dated data on the absolute altitudes of the sea level during the transgression of the Laptev Sea were used (Bauch et al., 2001). The transgression rate was not the same. The shortest episodes (400 and 375 years) of positive temperatures are determined for periods 15-11 and 10-9 ka respectively. The longest

intervals were 11–10, 9–5, and 5–0 Ka with 1000, 750 and >5000 years according to simple calculations. Special mathematical modeling for desalinated  coastal zones was not performed due to the relatively small area of their distribution    and the   specific nature of  salinity distribution over  the water column. Desalination of waters was taken into account directly when constructing a geocryological map.
* * *
**Comment**

Fig. 8: The map shows permafrost thickness, which can clearly result as output from 1D numerical modelling. What conditions were applied to determine zonation of permafrost based on  continuity (continuous,discontinuous,sporadic)? I.e. how do conclusions about distribution result from 1D modelling? Caption: why are only "fragments" of the map shown? Why not present the reader with the whole map?

**Response**

→ A paleogeographic scenario was constructed, consisting of a series of paleotemperature curves (or scenarios), in which the main factors that formed the modern permafrost were taken into account. Among these factors were: zoning, sectorality, fluctuations in sea level and its depth, which determined the period of freezing during shelf drainage and the thawing period when it was in the flooded state, the period of glaciation and thickness of the ice sheet, geothermal gradient, ground composition and properties, area, within which the water temperature rose at the Holocene optimum etc. In accordance with the indicated curves, modeling was carried out, the results of which reflected the influence on the permafrost of all of the above factors. The whole map is being prepared for publication in the atlas

Referee #2

**Comments from the reviewer #2 and our responses:**

**Comment**
1. Plot the air temperature (ground surface temperature) for different regions for the last 125k years. As I understand there is a temperature zonation factor involved. What is it? How does the air temperature differ between various regions, subzones?

**Response**

→ The article presents graphs of mean annual ground temperatures over the past 125 kyr depending on paleogeographic events (shelf drying, flooding, glaciation, etc.) according to latitudinal zoning and meridional sectorality (Fig. 4,5,7). When specifying latitudinal zonality during periods of shelf drying, the authors followed differences in ground temperatures reflected on the Russian Geocryological Map (Yershov ed., 1991). The mean annual ground temperature is 4-6 °C lower in the NE of the region (north of Taimyr) adjacent to the Kara coast, than as for the SW (south-west of Yamal). When zoning the shelf (Table 1, Fig. 3), the southwest (68-71 ° N) and north-eastern (72-77 ° N) areas were distinguished. During the periods of drainage in the periglacial part of the shelf in MIS-2, the following values for the ground temperature were taken: -19 ° C for the north-eastern area, and -15 °C for the south-western one .

B: glacial domain.

Fig. 4. Scenarios of sea level fluctuations:
 for the 15-0 kyr .: A -periglacial domain, B- glacial domain; for the 125-0 kyr: C - periglacial domain.
1 - sea level curve; 2 - dated bottom sediment cores from Ob' Gulf, Yenisei Gulf, and adjacent offshore (Stein, et al., 2009) and Vilkitsky Strait (Levitan.et al., 2007) sites; 3 - onshore data (Romanenko, 2012);
* * *
**Comment**

3. Make a series of maps to show which areas were flooded, glaciated or exposed to air 10, 20, 40, 55, 70, 80 ky bp. It will help a reader. It is okay to take different times, e.g. middle of the MIS periods. The goal is to help future researchers to understand when a certain part of the shelf was exposed to the air. Right now, it is not clear. Also, change the y-labels in Figure 4 to "Sea level with respect to the present-day datum, m". Make values in Figure 4a negative. '

**Response**

→ The authors totally agree with the reviewer: the relevant illustrative material will help the reader to understand the content of the article. Therefore, the change in surface conditions

is illustrated by a series of cartographic schemes for the following moments: MIS 5e (130-120 kyr,  MIS 5d (117-110 kyr)  MIS 5c (110-105 kyr); MIS 5b (100-75 kyr); 5a (75-65 kyr); MIS -4 (kyr); MIS -3 (50-25kyr); MIS -2 (25-15 kyr). See Lines 281-283:

→ As for the Figure 4 (a,b,c) we have changed y-labels for «altitude» to have them the same for all three plots to make it easier to the reader and made values  values in Figure 4a negative . Please refer to our above response.

→ The series of cartographic schemes illustrating , the change in surface conditions for the 125 kyr history of the Kara sea shelf  is presented in the Fig 8

[Figure]

→

→ Fig. 8. The series of cartographic schemes illustrating , the change in surface conditions in the Kara Sea shelf  for the following moments : MIS 5e (130-120 kyr,  MIS 5d (117-110 kyr)  MIS 5c (110-105 kyr); MIS 5b (100-75 kyr); 5a (75-65 kyr); MIS -4 (kyr); MIS -3 (50-25kyr); MIS -2 (25-15 kyr).
* * *
**Comment**

4. How are the effects of salts taken into the account? Does the salt lower the freezing point depression? Do you take into the account unfrozen liquid pore water while freezing the saline water? How are the salt effects parameterized in the model? Explain and clarify in the paper.

**Response**

→ The freezing temperature equal to the freezing temperature of sea water ( - 1,8°C) was set for all types of soils for the modeling. This assumption was made due to the fact that all the marine sediments composing the Yamal Peninsula have the close values of the salinity to a depth of 300 m and more (Chuvilin et al., 2007).Very high degree of averaging over the properties was used for the modeling caused by the lack of data on the water area. The rare drilling data showed the salinization of sediments through the entire drilling depth. So there was not possible to take into account the salt diffusion, and the salinity did not vary with the depth. Because the modeling has evaluative nature a scheme with complete freezing (thawing) of moisture in the ground at the moving front of phase transitions was used. The content of unfrozen water in the sediments was taken into account by reducing the volumetric heat of phase transitions in the model by the value corresponding to the average content of unfrozen water in different types of rocks at negative temperatures typical of the process under study.

**Response**

→ Actually we have a related comment from the first reviewer about lack of information on the influence of control factors of permafrost dynamics ( section 4.2). Therefore, it seems to be more convenient to expand that section with four figures showing the dynamics of the Kara shelf permafrost (top and base) for the past 125 Kyr. The results obtained during modeling for the following aspects for the southwestern and northeastern areas are presented now. The figure is presented in the supplements showing the influence of zonation, heat flux, lithology and properties and the influence of sea depths. The link to the reference is located at the Line 306 in the section « 4.2. Distribution of permafrost, its thickness and depth to the permafrost table»

:

→ This pattern has several controls (Supplementary Figure 1):

[Figure]

→ Supplementary Figure 1. The influence of various environmental factors on the dynamics of shelf permafrost over the past 125 kyr  according to the results of mathematical modeling
a) the influence of  latitudinal climatic zonation and division into sectors : southwestern (SW) and northeastern (NE) shelf parts (the loam, 50 m isobaths, q=50 mW/m2)
b) the influence of heat flux: 50 mW/m2  and 75 mW/m2 (the loam, 50 m isobaths, SW)
c) the influence of  lithology and properties of deposits : sand and loam ( 50 m isobath, q=50 mW/m2, SW )
d) the influence of sea depths (bottom isobaths): 5 and 50 m (the sand, q=50 mW/m2, SW)

→ As for the salinization, the modeling was provided for the uniform salt concentrations as we described in the response to the previous comment.
* * *
**Comment**
6. The manuscript is understandable, but terminology is used unconventionally
**Response**

→ Thank you, we moderated the language focusing on precision and word choice according to the Multi-Language Glossary of Permafrost (Everdingen, Robert van, ed. 1998 revised January 2002. Multi-language glossary of permafrost and related ground-ice terms. Boulder, CO: National Snow and Ice Data Center/World Data Center for Glaciology). Some of the modifications are listed below:
«cold» is changed to «cooled» that means the cryotic unfrozen deposits known as marine cryopeg We still  have chosen «cooled», because in Russian literature the cryopeg always means an isolated  cryopeg, and this can create a lot of confusion for non-native speakers. Therefore we decided to add that description «the cryotic unfrozen deposits are «cooled» deposits (marine cryopeg)».See Lines 290-291:

→ the cryotic unfrozen deposits are «cooled» deposits (marine cryopeg).

→ As for the rocks, we replaced the «rock» for the «deposits» (instead of rock and sediments) or «ground» (ground temperature instead of rock temperature) wherever it was necessary. Besides we have modified «unfrozen» deposits to «thawed» and made some more changes

Author comments about major changes in the article not mentioned in the response to the Reviewers

1.      We cropped the Figures 2,3,6 and 9 (previous № 8): in the previous version of the article,  the map covered part of the Laptev Sea, in this version there is just the Kara Sea.

2.      Fig.2 :The site №12 was missed, we added it to the current version

3.      The numbers of the figures was changed caused by the additional figure

4.      We made several changes in the Reference list

[revised manuscript text omitted]

**Thickness of relict continuous submarine permafrost, m**

100 - 300

**Thickness of relict discontinuous and sporadic submarine permafrost, m**

Discontinuous and sporadic permafrost 0-100 m thick

Sporadic permafrost 0-100 m thick

**Cooled and thawed deposits**

Cooled deposits

Thawed deposits

**Depths to relict submarine permafrost**

+ + + Shallow (0-30 m)

× × Medium (25-50 m)

—100— Izobaths, m

Research region boundary

[Figure]

**Thickness of relict continuous submarine permafrost, m**

100-200   400-500
300-400   500-600

**Thickness of relict discontinuous and sporadic submarine permafrost, m**

Discontinuous and sporadic permafrost, 0-100 m thick
Discontinuous and sporadic permafrost, 100-300 m thick
Sporadic permafrost, 0-100 m thick

**Cold and unfrozen rocks**

Cold rocks
Thawed rocks

**Depths to relict submarine permafrost**

Shallow (0-30 m)
Medium (25-50 m or more)
Deep (50-100 m)

—200— Izobaths, m
——— Research region boundary
Land

[revised manuscript text omitted]

---

## Editor Decision (ED1)

[revised manuscript text omitted]

Carrying out such a procedure is necessary  to reduce the number of simulated thermal tasks to a reasonable limit. There are several approaches.

**A.** In the first case, we consider the uniform alternation of homogeneous layers with a relatively low thickness (relative to the total thickness of the permafrost). The linear interpolation is  performed in accordance with the percentage of the thickness of frozen ground obtained during modeling for two "pure" soils at a given moment.

There are 2 options here:

1.  the thicknesses of frozen layers in sand ($T_s$) and loam ($T_l$) (clay) sediments from the simulation in similar conditions (scenario, heat flux). :

[Figure]

$$T_{al} = T_l + n_s * (T_s - T_l), \text{(1)}$$

where $n_s$ is the relative content of sand layers in the section.

 can also be used to evaluate the thawing of permafrost from above, if this occurs in accordance with a specific scenario.

2. More complicated is the very common case when, for example, relict permafrost is preserved in sandy sediments within the water area, and as for  loam it thaws completely long before modern times.

In this case, we apply the following .

For a reference soil in which permafrost is not preserved up to the present (and this is always loam (clay) in the considered pair of reference soils), the last point of the existence of permafrost at the $\tau_{deg}$ (time of degradation) is found  the corresponding graph.  This point, in the absence of thawing from above, is located on the surface, otherwise, at a certain depth from the bottom $z_{deg}$, where the upper and lower fronts of thawing  in the loams meet.

At this point, according to the slope of curve representing the  movement of the base of permafrost,  the rate of thawing $v_{th}$ from below is determined   before the complete disappearance of the frozen layer.  Based on this rate of the upward movement of the lower boundary of the permafrost, the value of the potential thawing of loams from the moment $\tau_{deg}$ is calculated:

[Figure]

$$\xi_{lb} = v_{th} \times \tau_{deg} , \qquad\qquad (2)$$

and the estimated (fictitious) position of the modern lower permafrost boundary in the loamy rocks relative to the bottom surface will be equal to

$$T_{lb}^f = z_{deg} - \xi_{lb} \qquad\qquad (3)$$

The value $T_{lb}^f$ can be either positive (below the bottom) or negative (above the bottom).

Putting $T_{lb}^f$  in (1) instead of Tc allows to find the position of the lower boundary $T_{al}$ of frozen soils in a layered sandy-loamy section with a given relative content of sand layers  $(n_s)$. A negative value clearly indicates the complete degradation of permafrost so far even in the absence of thawing from above.

To obtain the thawing from above for the alternating layers of sediments, the following approximate approach is used. At the moment of the complete degradation of frozen loams $\tau_{deg}$, according to the graph of permafrost dynamics, the thawing value from above $\xi_{ub}(\tau_{deg})$ is found for sandy soil and the ratio

$p = z_{\text{deg}}/\xi_{ub} \,(\tau_{\text{deg}})$ is calculated. It is assumed that the indicated ratio of thawing depths from above for soils of different compositions at the time of permafrost degradation in loamy sediments is preserved up to the present. Then the potential thawing on top of the loams at the moment will be

$$\xi_{ub}^{m} = p \times \xi_{ub} \quad (4),$$

where $\xi_{ub}^{m}$ and $\xi_{ub}$ are the calculated depth of thawing from the top for  loams and the model depth of thawing from the top for  sand, accordingly, to the current time

The current position of the permafrost  under the sea floor $\xi_m$ for a layered section of bottom sediments with a given relative content of sand $(n_s)$ is found from a dependence of type (1), which in this case has the  form:

$$\xi_m = \xi_{ub}^{m} + n_s \times (\xi_{ub} - \xi_{ub}^{m}) \qquad (5)$$

 the residual thickness of the layered permafrost under the sea bottom

$$th_{res} = T_{al} - \xi_m \qquad (6)$$

If the value $th_{res} < 0$ then the relict permafrost do not persist at a specific point in the water area.

B. Extrapolation of simulation results of submarine permafrost in homogeneous soils into a two-layer geological section.

The case is considered when the thickness of the sediments is relatively small and bedrock lies from a shallow depth under the bottom. An approximate approach is known to estimate the freezing depth of a two-layer strata with known freezing depths of the rocks of the upper and lower layers , all other things being equal.

This approach is based on obvious . At zero thickness of the upper layer of sediments ($th_1=0$), freezing of a two-layer system is equal to freezing in the lower layer $\xi_{2l} = \xi_l$, and  a thickness of the upper layer equal to the depth of freezing of the upper layer ($th_1 = \xi_u$) freezing of the system is equal to freezing of the upper layer $\xi_{2l} = \xi_u$.

Intermediate freezing depths of this two-layered system for random values of the upper layer thickness are considered varying linearly between these extreme values, where the following relationship is obtained:

$$\xi_{2l} = \xi_l + \left(1 - \frac{\xi_l}{\xi_u}\right) \times th_1 \qquad (7)$$

It is clear that the limits of variation  of the upper layer  are limited – when $th_1 > \xi_u$ the system becomes single-layer. Equation (7) is valid as well for thawing of  permafrost for two-layer structure .

 also two cases of interpolation:

1. The modern thickness of permafrost in sediments $(Th_{sed})$ and bedrocks $(Th_{br})$, is obtained  the same  conditions (scenario, heat flow) from the simulation results.  the thickness of permafrost in a two-layer system

$$Th_{2l} = Th_{br} + \left(1 - \frac{Th_{br}}{Th_{sed}}\right) \times th_1$$

where $th_1$ is the thickness of the upper layer of sediments or the depth of the top of the bedrocks.

We consider the sediments as the strata of alternating sand and loam with a given ratio in the section. The extrapolation of modeling results of homogeneous reference soils to a layered stratum was considered in the previous section.

The same dependence can also be used to evaluate thawing of two-layer permafrost from above.

2. Often there is a situation when the relict permafrost is preserved in sediments but in rocky deposits it is completely degraded to the current moment. In this situation, the same approach is used as previously considered (see section A). Using data on the depth and period of complete thawing of permafrost in bedrocks as well as on the dynamics of the movement of thawing fronts immediately before the disappearance of frozen rocks , the calculated (fictitious) position of the boundaries of frozen rocks at the modern time is determined using the dependencies (2-4), and the prediction of thawing of frozen rock from above is carried out according to (4) using the dynamics of sandy permafrost.

Using the same methodology, the fictitious positions of the upper and lower boundaries of the permafrost in sediments are calculated - i.e. layered strata with a given relative sand content. Further, substituting the obtained fictitious positions of the permafrost boundaries in sediments and rocks in (8), we find the position of these boundaries in a two-layer section. The current residual thickness of the permafrost is necessarily calculated as the difference in the depths of the positions of their lower and upper boundaries - a negative sign of this value indicates degradation of shelf permafrost.

[Figure]

**Supplementary Figure 1.** The influence of various environmental factors on the dynamics of shelf permafrost over the past 125 kyr  according to the results of mathematical modeling

a) the influence of  latitudinal climatic zonation and division into sectors : southwestern (SW) and northeastern (NE) shelf parts (the loam, 50 m isobaths, q=50 mW/m2)

b) the influence of heat flux: 50 mW/m2  and 75 mW/m2 (the loam, 50 m isobaths, SW)

c) the influence of  lithology and properties of deposits : sand and loam ( 50 m isobath, q=50 mW/m2, SW )

d) the influence of sea depths (bottom isobaths): 5 and 50 m (the sand, q=50 mW/m2, SW)

---

## Author Response (AR2)

We would like to thank the reviewers and the Editor for their careful re-review of our manuscript. Our point-by-point responses to all the comments and annotations are presented below. The comments from the reviewer are shown in black, and our responses in red. Where indicated, additions to the manuscript are shown in blue and deleted text is indicated by red strikethrough, and line numbers are based on our revised (non-track) manuscript. Besides, we added some information about changes in the paper that are not mentioned in the response to the Reviewers They are following:

1. We corrected the Figures 2, 3, 8 (previous № 6) replacing «км» to «km» at the scale bar and the Figures 5,6 (previous № 7) with «Kyr» corrected to «kyr»

2. We corrected the Fig. 7 (previous № 8) and it's caption to align with the text

==============================================================================

Referee #2

**Comments from the reviewer #2 and our responses:**

**Major comments:**

**Comment**
1. Lines 89-93: I am afraid the following is not easily reproducible by other researchers: "Then the modeling results were extrapolated to complex sections that comprise alternating reference lithologies or different combinations of relatively thick layers. We consider two cases to extrapolate our results: In the first case, we consider the uniform alternation of homogeneous layers with a relatively low thickness (relative to the total thickness of the permafrost). The linear interpolation is simply performed in accordance with the percentage of the thickness of frozen ground obtained during modeling for two —pure soils at a given moment." Please make a more precise and explicit description and verification of the proposed procedure. Details could be added to the supplemental material

**Response**
→ We added the detailed description of the method we used to extrapolate the results of the modeling to the different kinds of sections
→ Lines 101-102:
→ Therefore, as a first approximation, this change is linear (which is not entirely true) and a simple linear interpolation formula can be obtained (Supplement 1).
→

**Supplement 1.** Extrapolation of the modeling results of permafrost thickness to real geological sections.
Carrying out such a procedure is necessary to reduce the number of simulated thermal tasks to a reasonable limit. There are several approaches.
**A.** In the first case, we consider the uniform alternation of homogeneous layers with a relatively low thickness (relative to the total thickness of the permafrost). The linear interpolation is simply performed in accordance with the percentage of the thickness of frozen ground obtained during modeling for two "pure" soils at a given moment.
There are 2 options here:
1. We have the thicknesses of frozen layers in sand ($T_s$) and loam ($T_l$) (clay) sediments from the simulation in similar conditions (scenario, heat flux). Then the formula will be:
$$T_{al} = T_l + n_s * (T_s - T_l), (1)$$
where $n_s$ is the relative content of sand layers in the section

The same formula can also be used to evaluate the thawing of permafrost from above, if this occurs in accordance with a specific scenario

2. More complicated is the very common case when, for example, relict permafrost is preserved in sandy sediments within the water area, and in loam it thaws completely long before modern times.

In this case, we apply the following approximate approach.

For a reference soil in which frozen soils are not preserved up to the present (and this is always loam (clay) in the considered pair of reference soils), the last point of the existence of permafrost at the $\tau_{\text{deg}}$ (time of degradation) is found at the corresponding graph. This point, in the absence of thawing from above, is located on the surface, otherwise, at a certain depth from the bottom $z_{\text{deg}}$, where the upper and lower fronts of thawing of permafrost in the loams meet.

At this point, according to the slope of curve representing the movement of the base of permafrost, the rate of thawing $v_{\text{th}}$ from below is determined right before the complete disappearance of the frozen layer. Based on this rate of the upward movement of the lower boundary of the permafrost, the value of the potential thawing of loams from the moment $\tau_{\text{deg}}$ is calculated:

$$\xi_{lb} = v_{\text{th}} \times \tau_{\text{deg}} \text{ , (2)}$$

and the estimated (fictitious) position of the modern lower permafrost boundary in the loamy rocks relative to the bottom surface will be equal to

$$T_{lb}^f = z_{\text{deg}} - \xi_{lb} \text{ , (3)}$$

The value $T_{lb}^f$ can be either positive (below the bottom) or negative (above the bottom).

Putting $T_{lb}^f$ in (1) instead of Tc allows to find the position of the lower boundary $T_{al}$ of frozen soils in a layered sandy-loamy section with a given relative content of sand layers ($n_s$). A negative value clearly indicates the complete degradation of permafrost so far even in the absence of thawing from above.

To obtain the thawing from above for the alternating layers of sediments, the following approximate approach is used. At the moment of the complete degradation of frozen loams $\tau_{\text{deg}}$, according to the graph of permafrost dynamics, the thawing value from above $\xi_{ub}$ ($\tau_{\text{deg}}$) is found for sandy soil and the ratio

$p = z_{\text{deg}}/\xi_{ub}$ ($\tau_{\text{deg}}$) is calculated. It is assumed that the indicated ratio of thawing depths from above for soils of different compositions at the time of permafrost degradation in loamy sediments is preserved up to the present. Then the potential thawing on top of the loams at the moment will be

$$\xi_{ub}^m = p \times \xi_{ub} \text{ (4),}$$

where $\xi_{ub}^m$ and $\xi_{ub}$ are the calculated depth of thawing from the top for the loams and the model depth of thawing from the top for the sand accordingly, to the current time

The current position of the permafrost top under the sea floor $\xi_m$ for a layered section of bottom sediments with a given relative content of sand ($n_s$) is found from a dependence of type (1), which in this case has the next form:

$$\xi_m = \xi_{ub}^m + n_s \times (\xi_{ub} - \xi_{ub}^m) \text{ (5)}$$

Next, the residual thickness of the layered permafrost under the sea bottom should be found.

$$th_{res} = T_{al} - \xi_m \text{ (6)}$$

If the value $th_{res} < 0$ then the relict permafrost do not persist at a specific point in the water area.

B. Extrapolation of simulation results of submarine permafrost in homogeneous soils into a two-layer geological section.

The case is considered when the thickness of the sediments is relatively small and bedrock lies from a shallow depth under the bottom. An approximate approach is known to estimate the freezing depth of a two-layer strate with known freezing depths of the rocks of the upper and lower layers separately, all other things being equal.

This approach is based on obvious points. At zero thickness of the upper layer of sediments (th₁=0), freezing of a two-layer system is equal to freezing in the lower layer $\xi_{2l} = \xi_l$, and at a thickness of the upper layer equal to the depth of freezing of the upper layer (th₁ $= \xi_u$) freezing of the system is equal to freezing of the upper layer $\xi_{2l} = \xi_u$.

Intermediate freezing depths of this two-layered system for random values of the upper layer thickness are considered varying linearly between these extreme values, where the following relationship is obtained.

$$\xi_{2l} = \xi_l + (1 - \frac{\xi_l}{\xi_u}) \times th_1 \quad (7)$$

It is clear that the limits of variation in the thickness of the upper layer of ground are limited – when th₁ $> \xi_u$ the system becomes single-layer. Equation (7) is valid as well for thawing of the permafrost for two-layer structure section.

There are also two cases of interpolation.

1. The modern thickness of permafrost in sediments ($Th_{sed}$) and bedrocks ($Th_{br}$), is obtained under the same other conditions (scenario, heat flow) from the simulation results. Then the formula for determining the thickness of permafrost in a two-layer system will obviously have the next form:

$$Th_{2l} = Th_{br} + \left(1 - \frac{Th_{br}}{Th_{sed}}\right) \times th_1.$$

where $th_1$ is the thickness of the upper layer of sediments or the depth of the top of the bedrocks. We consider the sediments as the strata of alternating sand and loam with a given ratio in the section. The extrapolation of modeling results of homogeneous reference soils to a layered stratum was considered in the previous section.

The same dependence can also be used to evaluate thawing of two-layer permafrost from above.

2. Often there is a situation when the relict permafrost is preserved in sediments but in rocky deposits it is completely degraded to the current moment. In this situation, the same approach is used as previously considered (see section A). Using data on the depth and period of complete thawing of permafrost in bedrocks as well as on the dynamics of the movement of thawing fronts immediately before the disappearance of frozen rocks, the calculated (fictitious) position of the boundaries of frozen rocks at the modern time is determined using the dependencies (2-4), and the prediction of thawing of frozen rock from above is carried out according to (4) using the dynamics of sandy permafrost.

Using the same methodology, the fictitious positions of the upper and lower boundaries of the permafrost in sediments are calculated - i.e. layered strata with a given relative sand content. Further, substituting the obtained fictitious positions of the permafrost boundaries in sediments and rocks in (8), we find the position of these boundaries in a two-layer section. The current residual thickness of the permafrost is necessarily calculated as the difference in the depths of the positions of their lower and upper boundaries - a negative sign of this value indicates degradation of shelf permafrost.

**Comment**
2. 2. I could not find where the initial ground temperature distribution is given in the manuscript. The newly provided supplement indicates that the ground was thawed (no permafrost). Why was it assumed so? How would the modeling results differ if some permafrost was preserved 120kya?

**Response**
→    The initial distribution of ground temperatures is presented not only at the added figure. It is reflected in fig. 5 and 7 (current № 6), as a result of the existence of a marine warm-water basin transgressing to the continental coastal plains in MIS-5e (Map of Quaternary Formations of the Russian Federation (Zastrozhnov et al.), 2014; Astakhov and Nazarov, 2010; Gusev et al.,

2016). This basin began to form long before the MIS-5e optimum - about 140 kyr BP (Astakhov and Nazarov, 2010). The authors provided preleminary modeling for 200-117 kyr BP that showed the thawing of permafrost that had been formed before MIS-5e under the sea, which had existed for more than 20 kyr, as well as stabilization of the thermal field of ground. These results are consistent with the previously obtained ideas about the thawing of permafrost under the MIS-5e sea within the north of Western Siberia (Baulin, 1979; 1985). It was thicker permafrost since it was formed under more severe subaerial conditions and during longer period, but it was thawed. Therefore, there were no prerequisites for giving the existence of permafrost in MIS-5e. Still, if the existence of permafrost in MIS-5e was nevertheless given, it (existence) is unlikely to change the results obtained in this study. The frozen ground in MIS-5e and others formed before MIS-3 most likely had been thawed under the sea that existed in MIS-3 for 25 thousand years. The result would remain the same.

→        The next sentences were added (Lines 241-245):
→        The initial conditions are those given for the interglacial MIS-5e . There was a warm-water sea basin on the Kara shelf and adjacent lowlands  from 140 to 117 kyr (Fig. 5, 6) (Astakhov and Nazarov, 2010; Nazarov, 2011; Gusev et al., 2016a). Preliminary modeling for the interval MIS-6 - MIS-5e (200-117 kyr BP) showed that previously formed permafrost completely thawed under the sea during MIS-5e, which had existed for more than 20 kyr.The sequence of paleogeographic events in the form of a series of cartographic schemes is shown in Fig. 7.
→        The sentence was moved further from Line  229 to Line  260 (right after Fig.7)
→        .Altogether thirty such curves have been obtained, each based on four lithological patterns and two heat flux values.

Altogether thirty curves have been obtained, each based on four lithological patterns and two heat flux values.
→        The next sentence was added (Lines 260-262):
During transgressions in the Holocene An important role in the formation of the current permafrost is the last (Holocene) transgression of the sea. Therefore, it was set that, that each shelf part stayed for 400-2000 years in the coastal zone where bottom sediments were flooded with saline and warm near-bottom water. It is known that at sea depths from 2 to 7 m in the 1970s (Zhigarev, 1981), up to 10 m in the 2000s.

**Minor comments:**

**Comment**
1. Line 79: Please provide a reference to "the explicit two-layer scheme"
2. Line 79: Please provide a reference to "the balance method"
**Response**
We added the reference to the text and References (Lines 78-79 and Lines 642-643):
→        The explicit two-layer scheme is applied using the balance method (Pustovoit, G.P, 1999) and the enthalpy formulation of the problem.
→        Pustovoit, G.P.: Numerical solutions / Fundamentals of Geocryology. Part 5. (Engineering geocryology), edited by: Ershov, E. D., Moscow State University, Moscow, 41-55, 1999 (in Russian).

**Comment**
3. Please try to combine figures 5 and 7. I think it would be reasonable thing to do.

**Response**

→        The comment is right. We agree with the reviewer. The figure 7 is moved upper (it's № 6 now). The fig. 6 (it's № 8 now) is moved to the end of the Paleogeographic scenarios section. In current version the illustrations on paleotemperature curves follow one another. Besides, we made changes in the captions to drawings in order to have it more uniform.

→        Lines 247-249:

→        Fig. 5. Paleotemperature curves for the periglacial subaerial part of the shelf:

       1 = southwestern shelf part (SW), 80 m isobath; 2 = southwestern shelf part (SW), 5 m isobath;

       3 = northeastern shelf part (NE), 120 m isobath.

→        Lines 250-252

→        Fig.6.  Paleo-temperature curves for 50 m isobath :

       1 = periglacial subaerial northeastern shelf part (**NE**); 2 = ice that reached the bottom for 7-10 kyr, **G-1**); 3 = subaqual (beneath damlake) central shelf part, **C**;

**Comment**

4. Move figure 8 close to the introduction, prior to the development of the paleo scenarios.

**Response**

→        We moved Fig. 8 right after the Fig.5 and 6. It's current number is 7. Unfortunately, according to the logic of the article, we do not see the way of moving the figure closer to the introduction

**Comment**

5. Move figure 6 closer to the introduction as well. Modeling results should not be combined with the data driven figures. You provided a good justification for assuming a salt contaminated ground across the modeling domain in the responses to the reviewers. Please copy-paste it to the manuscript, in the current copy of the manuscript it is still a rather cryptic note.

**Response**

→        Unfortunately, according to the logic of the article, we do not see the way of moving the figure closer to the introduction

→        We added the text and the reference from the previous responses to the reviewers (Lines 103-114 and 539-541).

→        All reference rocks and sediments were considered saline with $D_s$ = 0.8-1.1% according to the concentration of pore saline solution corresponding to that of Kara sea bottom waters concentration (32-34 ‰). The freezing temperature equal to the freezing temperature of sea water ( - 1,8°C) was set for all types of soils for the modeling. This assumption was made due to the fact that all the marine sediments composing the Yamal Peninsula have the close values of the salinity to a depth of 300 m and more (Chuvilin et al., 2007).Very high degree of averaging over the properties was used for the modeling caused by the lack of data on the water area. The rare drilling data showed the salinization of sediments through the entire drilling depth. So there was not possible to take into account the salt diffusion, and the salinity did not vary with the depth. Because the modeling has evaluative nature a scheme with complete freezing (thawing) of moisture in the ground at the moving front of phase transitions was used. The content of unfrozen water in the sediments was taken into account by reducing the volumetric heat of phase transitions in the model by the value corresponding to the average content of unfrozen water in different types of rocks at negative temperatures typical of the process under study (Chuvilin et al., 2007).

→ Chuvilin, E.M., Perlova, E.V., Baranov, Yu.B., Kondakov, V.V., Osokin, A.B., Yakushev, V.S.: The structure and properties of cryolithozone sediments of the southern part of the Bovanenkovo gas condensate field. Geos, Moscow, 20, 2007 (in Russian).

**Comment**

6. Mark locations of the map that correspond to the scenarios in Figures 5 and 7

**Response**

→      In the figure captions there is an indication that the curves refer to certain isobaths. However, only the main ones (100, 200) are presented on the given fragment of the geocryological map. They give an idea of the location of the curves. Unfortunately, we do not see the possibility of otherwise expressing it. For greater clarity, we have added the 50m isobath to the map.

[Figure]

**Comment**

7. In Table 1, add years when the ice-dammed lake existed.

→ The dammed basins existed when the MIS-2 glacier existed. We have set it  from 25 to 15 kyr. We added this information to the Table 1

[revised manuscript text omitted]

Fig. 6̶7̶. P̶a̶l̶e̶o̶g̶e̶o̶g̶r̶a̶p̶h̶i̶c̶ ̶s̶c̶e̶n̶a̶r̶i̶o̶ Paleo-temperature curves for 50 m isobath ̶:̶f̶o̶r̶ ̶M̶I̶S̶-̶2̶ ̶w̶h̶i̶c̶h̶ ̶c̶o̶n̶t̶r̶o̶l̶l̶e̶d̶ ̶t̶h̶e̶ ̶p̶r̶e̶s̶e̶n̶t̶ ̶d̶i̶s̶t̶r̶i̶b̶u̶t̶i̶o̶n̶ ̶o̶f̶ ̶f̶r̶o̶z̶e̶n̶ ̶a̶n̶d̶ ̶c̶o̶o̶l̶e̶d̶ ̶g̶r̶o̶u̶n̶d̶.̶ ̶A̶d̶a̶p̶t̶e̶d̶ ̶f̶o̶r̶ ̶m̶o̶d̶e̶l̶i̶n̶g̶ ̶p̶u̶r̶p̶o̶s̶e̶s̶.̶ ̶A̶ ̶f̶r̶a̶g̶m̶e̶n̶t̶.̶
1 = periglacial subaerial northeastern shelf part (**NE**); 2 = ice that reached the bottom for 7-10 kyr, **G-1**); 3 = subaqual (beneath damlake) central shelf part, **C**;

255

[Figure]

MIS-5e (140-117 kyr BP)

MIS-5d (117-110 kyr BP)

MIS-5c (110-100 kyr BP)

MIS-5b (100-77 kyr BP)

MIS-5a (77-65 kyr BP)

MIS-4 (65-50 kyr BP)

MIS-3 (50-25 kyr BP)

MIS-2 (25-15 kyr BP)

[revised manuscript text omitted]

---

## Author Response (AR3)

**Dear Editor, we would like to thank you for careful reading and remarks, we corrected the manuscript and supplement according to them. Where indicated, additions to the manuscript are shown in blue and deleted text is indicated by red strikethrough.**

[revised manuscript text omitted]

**Supplement 1. Extrapolation of the modeling results of permafrost thickness to real geological sections.**

Carrying out such a procedure is necessary to reduce the number of thermal simulations to a reasonable limit. There are several approaches.

**A.** In the first case, we consider the uniform alternation of homogeneous layers with a relatively small thickness (relative to the total thickness of the permafrost). The simple linear interpolation is performed in accordance with the percentage of the thickness of frozen ground obtained during modeling for two "pure" soils at a given moment.

There are 2 options here:

1. Using the thicknesses of frozen layers in sand ($T_s$) and loam ($T_l$) (clayey) sediments from the simulation in similar conditions (scenario, heat flux), we can use the equation:

$$T_{al} = T_l + n_s * (T_s - T_l), \quad (1)$$

where $n_s$ is the relative content of sand layers in the section( p.u.), $T_{al}$ is the thickness of permafrost for alternating layers (m), Ts and Tl are the thickness of permafrost for sand and loam (clayey) soils, respectively.

Equation (1) can also be used to evaluate the thawing of permafrost from above, if this occurs in accordance with a specific scenario.

2. More complicated is the very common case when, for example, relict permafrost is preserved in sandy sediments within the water area, and, as for loam, it thaws completely long before modern times.

In this case, we apply the following approximation.

For a reference soil in which permafrost is not preserved up to the present (and this is always loam (clay) in the considered pair of reference soils), the last point of the existence of permafrost at the $\tau_{deg}$ (time of degradation) is found within the corresponding simulation. This point, in the absence of thawing from above, is located on the surface. Otherwise, it will be located at a certain depth from the bottom $z_{deg}$, where the upper and lower fronts of permafrost thawing in the loams meet.

At this point, according to the slope of the curve representing the movement of the base of permafrost, the rate of thawing $v_{th}$ from below is determined (exactly before the complete disappearance of the frozen layer). Based on this rate of the upward movement of the lower boundary of the permafrost, the value of the potential thawing of loams from the moment $\tau_{deg}$ onward is calculated:

$$\xi_{lb} = v_{th} \times \tau_{deg} , \quad (2)$$

where $\xi_{lb}$ is the potential thawing of loams (m), $v_{th}$ is the rate of thawing from below ( m*yr$^{-1}$), $\tau_{deg}$ is the time of degradation (yr).

and the estimated (fictitious) position of the modern lower permafrost boundary in the loam relative to the bottom surface will be equal to

$$T_{lb}^f = z_{deg} - \xi_{lb} \quad (3)$$

where $T_{lb}^f$ is the estimated (fictitious) position of the modern lower permafrost boundary in the loam (m), $z_{deg}$ is the depth corresponding to the $\tau_{deg}$(m).

Hereby, the value $T_{lb}^f$ can be either positive (below the bottom) or negative (above the bottom).

Putting $T_{lb}^f$ in (1) instead of $T_l$ allows to find the position of the lower boundary $T_{al}$ of frozen soils in a layered sandy-loamy section with a given relative content of sand layers ($n_s$). A negative value clearly indicates the complete degradation of permafrost so far, even in the absence of thawing from above.

To obtain the thawing from above for the alternating layers of sediments, the following approximate approach is used. At the moment of the complete degradation of frozen loams $\tau_{deg}$, according to the simulation, the thawing value from above $\xi_{ub}$ ($\tau_{deg}$) is found for sandy soil and the ratio $p = z_{deg}/\xi_{ub}$ ($\tau_{deg}$) is calculated. It is assumed that the indicated ratio of thawing depths from above for soils of different compositions at the time of permafrost degradation in loamy sediments is preserved up to the present. Then the potential thawing on top of the loams at the moment will be

$$\xi_{ub}^m = p \times \xi_{ub} \quad (4),$$

where $\xi_{ub}^m$ and $\xi_{ub}$ are the calculated depths of thawing from the top for the loams and the model depth of thawing from the top for  sand respectively at the modern times (m), $p$ is the ratio of certain depth from the bottom $z_{deg}$, where the upper and lower fronts of permafrost thawing in the loams meet and the thawing depth for the sand for the moment $\tau_{deg}$ of complete degradation of frozen loams (p.u.)

The  modern position of the permafrost table under the sea floor $\xi_m$ for a layered section of bottom sediments with a given relative content of sand ($n_s$) is found from a dependence of type (1), which in this case has the  form:

$$\xi_m = \xi_{ub}^m + n_s \times (\xi_{ub} - \xi_{ub}^m) \quad (5)$$

where $\xi_m$ is the modern depth of the permafrost table (m), other parameters of the expression are given above/

The residual thickness of the layered permafrost under the sea bottom can then be expressed as:.

$$th_{res} = T_{al} - \xi_m \quad (6).$$

where $th_{res}$ is the residual thickness of the layered permafrost under the sea bottom (m).

If the value $th_{res} < 0$ then the relict permafrost do not persist at a specific point in the water area.

**B. Extrapolation of simulation results of submarine permafrost in homogeneous soils into a two-layer geological section.**

The case is considered when the thickness of the sediments is relatively small and bedrock is present at shallow depth  under the bottom. An approximate approach is known to estimate the freezing depth of a two-layer strata with known individual freezing depths of the ground of the upper and lower layers, all other things being equal.

This approach is based on the two extreme cases. At zero thickness of the upper layer of sediments (th$_1$=0), freezing of a two-layer system is equal to freezing in the lower layer $\xi_{2l} = \xi_l$ , and  for a thickness of the upper layer equal to the depth of freezing of the upper layer (th$_1 = \xi_u$ ), the freezing of the system is equal to freezing of the upper layer $\xi_{2l} = \xi_u$.

Intermediate freezing depths of this two-layered system for random values of the upper layer thickness are considered varying linearly between these two extreme values, where the following relationship is obtained:

$$\xi_{2l} = \xi_l + (1 - \frac{\xi_l}{\xi_u}) \times th_1 \quad (7)$$

where $\xi_{2l}$ is the permafrost thickness of the two-layer strata (m), $\xi_l$ and $\xi_u$ is the permafrost thickness of the lower and upper layers respectively (m), $th_1$ is the thickness of the upper layer.

It is clear that the limits of  thickness variation of the upper layer  are limited – when $th_1 > \xi_u$ the system becomes single-layer. Equation (7) is valid as well for the thawing of  permafrost for two-layer  structures.

 We adress also two cases of interpolation:

1. The modern thickness of permafrost in sediments ($Th_{sed}$) and bedrocks ($Th_{br}$) is obtained  for the same  conditions (scenario, heat flow) from the simulation results. the thickness of permafrost in a two-layer system can then be expressed as :

$$Th_{2l} = Th_{br} + \left(1 - \frac{Th_{br}}{Th_{sed}}\right) \times th_1 \ \underline{(8),}$$

 where $T_{2l}$ is the thickness of of permafrost in a two-layer strata (m), $T_{sed}$ and $T_{br}$ is the modern thickness of permafrost in sediments and bedrocks, $th_1$ is the thickness of the upper layer of sediments equal to the depth of the top of the bedrocks.

We consider the sediments as the strata of alternating sand and loam with a given ratio in the section. The extrapolation of modeling results of homogeneous reference soils to a layered stratum was considered in the previous section.

The same dependence can also be used to evaluate the thawing of two-layer permafrost from above.

2. There exist often  a situation when  relict permafrost is preserved in sediments, but  it is completely degraded today in bedrocks according to simulation . In this situation, the same approach is used as previously considered as in section A is applied.  Using data on the depth and period of complete thawing of permafrost in bedrock as well as on the dynamics of the movement of the thawing front immediately before the complete disappearance of frozen rock, the calculated (fictitious) position of the boundaries of frozen rocks at  modern times is determined using the  equations (2-4), and the prediction of thawing of frozen rock from above is carried out according to equation (4) using the dynamics of sandy permafrost.

Using the same methodology, the fictitious positions of the upper and lower boundaries of the permafrost in sediments are calculated - i.e. layered strata with a given relative sand content. Further, substituting the obtained fictitious positions of the permafrost boundaries in sediments and rocks in equation (8), we find the position of these boundaries in a two-layer section. The current residual thickness of the permafrost is necessarily calculated as the difference in the depths of the positions of their lower and upper boundaries - a negative sign of this value indicates degradation of shelf permafrost.

[Figure]

**Supplementary Figure 1.** The influence of various environmental factors on the dynamics of shelf permafrost over the past 125 kyr according to the results of mathematical modeling

a) the influence of latitudinal climatic zonation and division into sectors : southwestern (SW) and northeastern (NE) shelf parts (the loam, 50 m isobaths, q=50 mW/m2)

b) the influence of heat flux: 50 mW/m2 and 75 mW/m2 (the loam, 50 m isobaths, SW)

c) the influence of lithology and properties of deposits : sand and loam ( 50 m isobath, q=50 mW/m2, SW )

d) the influence of sea depths (bottom isobaths): 5 and 50 m (the sand, q=50 mW/m2, SW)